# EP300 deficiency leads to chronic replication stress mediated by defective replication fork protection

Angelica Barreto-Galvez[1,12], Mrunmai Niljikar [1,12], Julia Elizabeth Gagliardi[1], Carolina Plasencia Guzman[1], Ranran Zhang[1], Vasudha Kumar[1], Aastha Juwarwala [1], Archana Pradeep[1], Ankit Saxena[1], Cristina Montagna[1,2], Priya Mittal [3], Jeannine Gerhardt[4], Bing Xia [5], Jian Cao[1,6], Keisuke Kataoka[7], Adam David Durbin [8], Jun Qi [9], B. Hilda Ye [10] & Advaitha Madireddy [1,11] ✉

Mutations in the global transcriptional activator EP300/KAT3B are being reported in aggressive malignancies. However, the mechanistic contribution of EP300 dysregulation to cancer is currently unknown. While EP300 has been implicated in regulating cell cycle and DNA replication, the role of EP300 in maintaining replication fork integrity has not been studied. Here, using EP300-mutated adult T-cell leukemia/lymphoma cells and an EP300-selective degrader, we reveal that EP300 loss leads to pronounced dysregulations in DNA replication dynamics and persistent genomic instability. Aberrant DNA replication in EP300-mutated cells is characterized by elevated replication origin firing due to replisome pausing. EP300 deficiency results in a prominent defect in fork protection resulting in the accumulation of single-stranded DNA gaps. Importantly, we find that the loss of EP300 results in decreased expression of BRCA2 protein leading to sensitivity to treatments that are cytotoxic to BRCA-deficient cancers. Overall, we demonstrate that EP300-mutated cells recapitulate features of BRCA-deficient cancers.

Loss-of-function mutations in EP300/KAT3B and its close homolog, CBP/KAT3A, have been implicated in the pathogenesis of many hematologic malignancies[1]. EP300/CBP are global transcriptional coactivators that catalyze the addition of acetyl groups to lysine residues on histones (HAT activity) and non-histone proteins[2–5] (KAT activity). Reports suggest that the independent inhibition of EP300 in human cells results in the differential expression of genes involved in regulating the cell cycle, DNA replication and DNA repair. While CBP

downregulation results in changes to genes involved in antigen presentation, and terminal B-cell differentiation.

In this study, we sought to characterize the independent role of EP300 in DNA replication integrity by studying patient-derived North American ATLL cell lines, that carry loss-of-function somatic mutations in *EP300* (but not *CBP*)[6]. In a recent retrospective analysis of a single-center cohort of Caribbean/American ATLL (NA-ATLL), the median overall survival (OS) was only 6.9 months[7–10], worse than the OS

[1]Rutgers Cancer Institute of New Jersey, New Brunswick, NJ, USA. [2]Department of Genetics, Albert Einstein College of Medicine, Bronx, NY, USA. [3]Center of Excellence in Neuro-Oncology Sciences (CENOS), St. Jude Children's Research Hospital, Memphis, TN, USA. [4]Department of Obstetrics and Gynecology, Weill Cornell Medicine, New York, NY, USA. [5]Department of Radiation Oncology, Rutgers University, New Brunswick, NJ, USA. [6]Department of Medicine, Robert Wood Johnson Medical School, Rutgers University, New Brunswick, NJ, USA. [7]Division of Molecular Oncology, National Cancer Center Research Institute, Tokyo, Japan. [8]Department of Oncology, St. Jude Comprehensive Cancer Center, Memphis, TN, USA. [9]Department of Cancer Biology, Dana Farber Cancer Institute, Harvard Medical School, Boston, MA, USA. [10]Department of Cell Biology, Albert Einstein College of Medicine, Bronx, NY, USA. [11]Department of Pediatrics Hematology/Oncology, Robert Wood Johnson Medical School, Rutgers University, New Brunswick, NJ, USA. [12]These authors contributed equally: Angelica Barreto-Galvez, Mrunmai Niljikar. ✉e-mail: Advaitha.madireddy@rutgers.edu

outcome in the largest ATLL cohort from Japan (J-ATLL) of ~1 year[11]. Further analysis revealed that 20% of NA-ATLL patients had mutations in the EP300 gene, and that ATLL patients with epigenetic mutations had a worse prognosis as compared to those without these mutations. Several patient-derived cell lines[12] were subsequently developed that differ in EP300 mutation status. These cell line models provide a powerful tool to examine genotype-to-phenotype correlations associated with EP300 deficiency. To directly establish causal connections, we also took advantage of a highly selective EP300-specific PROTAC degrader that does not target CPB (a common problem with commercial EP300 inhibitors) and has limited toxicity in vivo[13].

Importantly, almost all functional studies, in the literature, either examine EP300 and CBP together or assess the importance of EP300 to DNA repair since studies look at response to DNA damaging agents such as UV irradiation or ionizing radiation. For example, histone acetylation at double strand breaks (DSB) by EP300/CBP has been shown to DNA repair proteins to chromatin[14–16]. In addition, numerous studies show that EP300/CBP-mediated acetylation of proteins involved in DNA replication/repair can stimulate or inhibit their activities in reconstituted systems. These studies have revealed that EP300/CBP can directly interact with PCNA in vitro to stimulate DNA synthesis after UV irradiation[17], they can associate with ATR checkpoint signaling kinase[18] and through their acetylation role (KAT), regulate the Fen1 endonuclease to help with Okazaki fragment maturation after UV irradiation[19], DNA2 endonuclease[20], DNA polymerase beta in base excision repair[21], DNA glycosylases for base mismatch repair[22,23], the Werner helicase after UV induced damage[24] and numerous reports suggest their importance to homologous recombination repair[25]. Despite the suggested role for EP300 in DNA replication, it is not known whether inactivating mutations in EP300 spontaneously induce endogenous replication stress. Importantly, functional studies analyzing how these EP300Mut cells overcome replication fork stalling in the presence of replicative inhibitors such as aphidicolin and hydroxyurea have not been conducted.

DNA replication integrity refers to the faithful and timely duplication of the genome to maintain genome stability. Replication stress (endogenous or induced), the biggest threat to genome stability, can stall the replisome and if not resolved efficiently, can lead to genomic instability. Upon stalling, human cells have evolved several mechanisms to stabilize and restart replication forks to prevent fork collapse leading to DNA breaks. Cells activate the ATR/Chk1 checkpoint kinase signaling pathway in response to replication stress. Phosphorylation of ATR, Chk1, RPA, and histone H2AX then ensures that DNA replication stalls in order to resolve stalled replication forks before replication can continue. Fork restart is then mediated by a complex set of steps that involve fork reversal, fork stabilization by the RAD51 recombinase, and downstream mediators such as BRCA2/BRCA1/PALB2 and fork restoration by error-free recombination-based or error-prone POLD3 mediated-mechanisms. While EP300 has been implicated in transcriptionally activating BRCA1 and RAD51 to promote homologous recombination[25,26], the functional impact of EP300 downregulation on replication fork stability in response to replication stress has not been evaluated.

To address EP300's role in DNA replication, we analyzed replication integrity and mechanisms driving genomic instability in EP300-mutated cells. We found that EP300-mutated cells display a protracted cell cycle, extensive genomic instability, dysregulations in DNA replication and repair dynamics. We show that perturbed DNA replication in EP300-mutated ATLL cells is characterized by elevated replication origin firing, increased replisome pausing, and nucleolytic degradation of nascent DNA at collapsed forks. This in turn results in the persistence of single-stranded DNA (ssDNA) genome-wide in EP300-deficient NA-ATLL cells. Inhibition of nucleases such as Mre11 and DNA2 rescued the ssDNA accumulation, indicating a dysregulation in downstream mechanisms that restrain nuclease activity at stalled forks. Notably, we found a distinct loss of BRCA2 protein expression in EP300Mut cells. Importantly, degradation of EP300 in EP300WT cells recapitulated the phenotypes observed in EP300Mut patient cells. These S-phase abnormalities result in under-replicated DNA in G2/M, instigate mitotic DNA synthesis, and are associated with mitotic segregation defects, elevated micronuclei, and inherited DNA lesions in G1-phase. Taken together, we show that EP300Mut human cells recapitulate features of BRCA-deficient cells and are susceptible to inhibitors that are in clinical use or preclinical testing for BRCA-deficient cancers.

## Results

**EP300 loss alters histone acetylation and transcriptional programs.** In this study, four North American Adult T-cell leukemia/lymphoma (NA-ATLL) patient-derived cell lines were analyzed, which included two EP300 wildtype (NA1, NA4) and two EP300 mutants (NA2 and NA3). In addition, we included two Japanese ATLL lines as a control cohort, which included a EP300 wildtype (J2) and EP300 mutant (J1) (Supplemental Table 1). An immunoblot analyzing EP300 and CBP protein expression in these cell lines showed a prominent EP300 band in the EP300WT cells. In contrast, the EP300 mutant cells showed little to no EP300 protein expression. All cell lines irrespective of the EP300 protein expression status had strong and comparable CBP protein expression (Supplementary Fig. 1A). In addition, gene expression analysis revealed a clear separation in transcription programs between EP300Mut and EP300WT cells (Supplementary Fig. 1B). To establish a causal role for EP300 in the phenotypes observed, we utilized a highly selective EP300-specific PROTAC degrader, JQAD1, that does not target CPB[13]. EP300WT cells treated with negative control (S,S) stereoisomer were used as a control. To begin, we standardized JQAD1 dosage/duration of treatments in EP300WT NA-ATLL cells. While a high dose of JQAD1 (10 uM) was effective in degrading EP300 as early as 24 h, a lower dose of 500 nM JQAD1 degraded EP300 by 72 h (Supplementary Fig. 1C). We decided to proceed with the lower dose for both EP300WT NA-ATLL lines (Supplementary Fig. 1D). Acetylation on histone H3 lysine 27 (H3K27ac) is catalyzed by EP300 and its paralogue CBP[27]. To functionally test the effect of EP300 degradation on H3K27 acetylation in EP300WT cells, we carried out immunoblotting analysis of protein extracts from EP300WT ATLL cells lines (NA1 and NA4) treated with 500 nM JQAD1. JQAD1-mediated EP300 degradation resulted in a significant decrease in the H3K27ac mark as early as 3 days and beyond at 5 and 7 days post treatment (Supplementary Fig. 1E). Comparison of gene expression profiles between a EP300WT (NA1) NA-ATLL cell line and its JQAD1 treated pair revealed a distinct separation in expression profiles providing functional validation of the efficacy of JQAD1 in degrading EP300 (Supplementary Fig. 1F).

**EP300-mutated cells have delayed S-phase kinetics.** To assess the effect of EP300 deficiency on cell cycle kinetics, we first calculated cell doubling times in EP300Mut, EP300WT, and EP300WT + JQAD1 cell lines. We included two non-ATLL human cell lines (B-lymph and JURKAT) for this analysis. The results revealed that while EP300WT cells take ~20 h, EP300Mut cells take twice as long ( ~ 40 h) to divide (Fig. 1A, B). To identify the cell cycle phase responsible for this, we monitored cell cycle progression by co-staining for 5-ethynyl-2′-deoxyuridine (EdU) and propidium iodide (PI) and quantified the percentage of cells in G0/G1, S, and G2/M phases by flow cytometry. The results revealed a 2-fold increase in the S-phase fraction of EP300Mut cells as compared to EP300WT NA-ATLL cells (Fig. 1C), indicating that slower cell cycle kinetics were due to slower progression of EP300Mut cells through the S-phase. To establish that S-phase arrest was indeed EP300-associated, we evaluated cell cycle profiles by PI staining in EP300WT and Mut cells after 3, 5 and 7 days of JQAD1 treatment. Results showed that EP300WT cells had a 2-fold increase in the S-phase fraction of cells upon EP300 loss (Supplementary Fig. 1G). This increase started as early as day 3 and persisted till days 5 and 7. The S-phase arrest observed

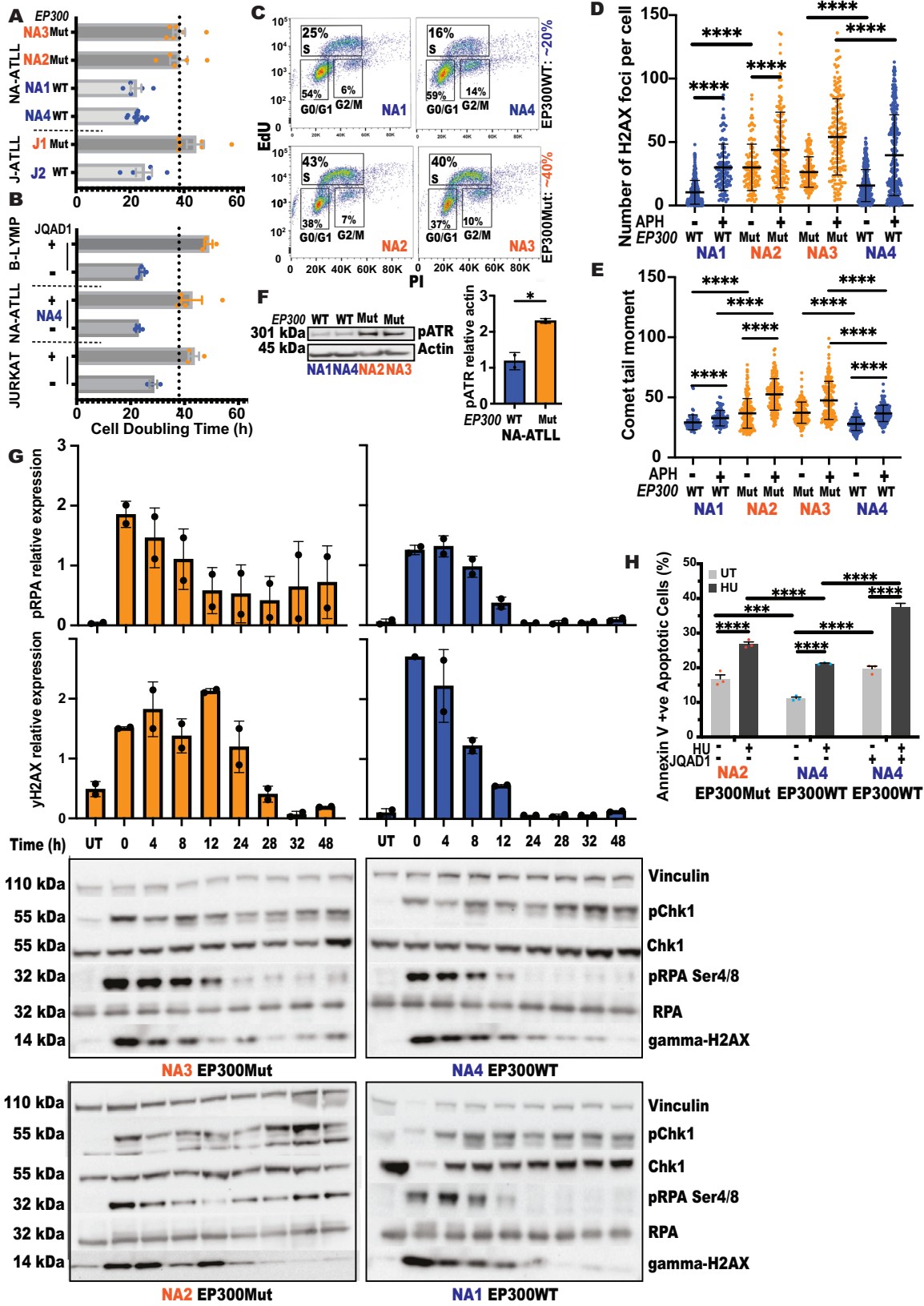

upon EP300 loss was similar to that observed inherently in EP300Mut NA-ATLL cells with or without JQAD1 treatment (Supplementary Fig. 1G). These results collectively point to increased replication stress in the absence of EP300.

EP300 deficient cells have spontaneous DNA damage and increased sensitivity to exogenous replication stress. To determine whether slower S-phase progression was due to accumulating DNA damage, we next measured the overall magnitude of genomic instability by immunofluorescence staining (IF) for phosphorylation of histone variant 2 A (pH2AX). This analysis revealed a spontaneous increase in pH2AX foci per cell in EP300Mut NA-ATLL cells, as compared to wildtype cells (Fig. 1D). Low dose aphidicolin (APH; replicative

**Fig. 1 | *EP300*-deficient cells have aberrant cell cycle dynamics, spontaneous DNA damage, increased sensitivity to replication stress, and persistent checkpoint activation. A**, **B** Cell doubling time calculated every 48 h from three independent cultures of untreated (**A**) and 500 nM JQAD1-treated (**B**) EP300WT ATLL or non-ATLL B-lymphocyte cells; for all experiments, data are presented as mean values +/− SEM; for (**A**), *N* = 5 and for (**B**), *N* = 3 (**C**) Flow cytometry-based cell cycle analysis of untreated EP300WT (top) and EP300Mut (bottom) NA-ATLL cells. Cell cycle progression was monitored by co-staining for 5-ethynyl-2′-deoxyuridine (EdU) and propidium iodide (PI). **D** Analysis of the number of phospho- H2AX (phosphorylation of Histone H2AX at Ser139) foci per cell nuclei in EP300Mut and EP300WT NA-ATLL cells treated with 0.4uM Aphidicolin (APH) overnight (O/N), *N* = 3. **E** Measurement of DNA single-strand breaks by alkaline Comet assay in EP300Mut and EP300WT cells treated with 0.4uM APH (O/N). Comet tail lengths were measured using the OpenComet plugin as part of the ImageJ software, *N* = 3. **F** Left: Expression levels of phospho-ATR in EP300WT/Mut NA-ATLL cells in the absence of exogenous replication inhibition. Right: Relative protein expression of pATR in EP300Mut and WT NA-ATLL cells, N = 2. **G** Time course experiment to measure recovery of NA-ATLL EP300Mut/WT cells from 2 mM Hydroxyurea treatment (4 h) after release into drug-free media over the course of 48 hours. Cells were collected at eight time points (0, 4, 8, 12, 24, 28, 32, 48 hours) and expression levels of phospho-RPA Ser4/8 and phospho-Histone H2AX Ser139 were measured by western blotting. Expression levels of Vinculin were used as a loading control for phospho-H2AX, and pRPA was measured relative to RPA32, *N* = 2. **H** Quantification of % AnnexinV/PI +ve cells assessed by flow cytometry in EP300Mut, EP300WT NA-ATLL cells +/− 500nMJQAD1 (EP300-specific PROTAC degrader) and +/− 2 mM HU (4 h), *N* = 3. For all experiments, data are presented as mean values +/− SD. *P*-values were determined by a two-tailed Student T-test. The *p*-values are indicated as follows: * <0.03, ** <0.0021, *** <0.0002, **** <0.0001. N represents three experimental replicates from independent cultures of cells as indicated. Source data are provided as a Source Data file.

inhibitor) treatment exacerbated the damage in all cell lines, however, EP300Mut cells had significantly more DNA breaks than WT cells (Fig. 1D). While JQAD1-mediated degradation of EP300 in WT cells resulted in a spontaneous increase in DNA breaks, exposure of EP300Mut cells to JQAD1 did not show any further increase in the damage observed (Supplementary Fig. 1H). To establish the contribution of EP300 to damage induction, we released JQAD1 treated cells into drug-free media for 72 h and assessed pH2AX foci. The results showed a decrease in pH2AX foci numbers in the released cells comparable to the untreated WT levels (Supplementary Fig. 1I). No notable change was observed in EP300Mut cells (NA2, NA3) upon JQAD1 treatment or release. To confirm spontaneous and induced genomic instability at the single-cell level, we carried out the alkaline Comet assay[28]. Results revealed a significant increase in both spontaneous and APH induced comet tail movement in EP300Mut NA-ATLL cells (orange), compared to the EP300WT cells (blue; Fig. 1E). JQAD1 treatment in EP300WT but not the EP300Mut cells resulted in longer comet tail lengths as compared to untreated cells (Supplementary Fig. 1J). Taken together, these studies collectively show that EP300 deficiency leads to delayed cell cycle kinetics due to accumulating DNA damage.

EP300-mutated NA-ATLL cells have persistent checkpoint signaling associated with unresolved DNA damage. Since replicative difficulties during the S-phase result in activation of ATR-Chk1 signaling cascade[29], we next evaluated checkpoint activation by immunoblotting in EP300Mut/WT cells in the presence or absence of hydroxyurea (HU, replicative inhibitor). Analysis of ATR phosphorylation revealed a striking increase in spontaneous pATR in EP300Mut cells, as compared to wildtype cells (Fig. 1F). In addition, while inhibitor treatments induced robust Chk1 phosphorylation in all cell lines, a time course experiment revealed a defect in damage and checkpoint resolution in EP300Mut cells (Fig. 1G). To establish that replication arrest mediated by Chk1 phosphorylation is in response to replication perturbation, we assessed phosphorylation of RPA at Ser 4/8 (pRPA4/8), a marker of collapsed forks and ssDNA[30]. Again, while all cell lines activated pRPA4/8 in response to damage, EP300Mut cells hyperphosphorylated RPA, indicating increased collapsed forks and ssDNA in the absence of EP300. Importantly, while pRPA4/8 and pH2AX were resolved effectively by 24 h in EP300WT cells, they persisted in EP300Mut cells even at 48 h (Fig. 1G). These results show that spontaneous replication stress in the absence of EP300 is associated with ineffective replicative checkpoint resolution.

To assess the effect of inefficient checkpoint resolution on cell viability, we next measured the sensitivity of EP300 deficient cells to exogenous replication inhibition. We analyzed spontaneous and HU-induced apoptotic fractions by flow cytometry using Annexin V/PI staining. The results revealed a significant spontaneous increase in apoptosis in EP300Mut NA-ATLL cells and JQAD1-treated EP300WT cells, as compared to isogenic EP300WT cells (Fig. 1H). This effect was further exacerbated upon HU treatment. This,

combined with the extensive S-phase abnormalities, suggests a potential vulnerability of EP300Mut cells to replicative inhibitors.

EP300-mutated cells have aberrant replication dynamics at CFS. To understand the endogenous events driving or resulting from spontaneous replication stress in EP300Mut ATLL cells, we next investigated the DNA replication dynamics. Chromosomal instability in NA-ATLLL cells has been shown to cluster at difficult-to-replicate regions of our genome called common fragile sites (CFS)[31–33]. We are uniquely equipped to study locus-specific DNA replication using our powerful DNA replication assay, Single Molecule Analysis of Replicated DNA (SMARD; Supplemental Fig. 2A). Here, we used SMARD. To understand the effect of EP300 deficiency on in vivo DNA replication dynamics at common fragile site FRA6E (CFS-FRA6E; Fig. 2A), in pairs of EP300Mut/WT NA-ATLL and J-ATLL cells. Under normal conditions, the replication program at CFS-FRA6E in an EP300WT cell line is predominantly replicated by forks travelling in the 5′ to 3′ direction, and there are no major replication origins within the locus[34]. The absence of replication origins is a defining characteristic of CFS loci, and we have previously established that any origins observed are a stress response either due to the absence of an essential protein[35] or due to exposure to replicative inhibitors[36]. Replication dynamics at FRA6E in a J-ATLL or NA-ATLL cell line expressing wildtype EP300 revealed a replication program very similar to untreated B-lymphocyte FRA6E profiles previously characterized by our lab[34]. The FRA6E locus was replicated by forks primarily moving in the 5′ to 3′ direction with no prominent replication initiation sites detected with the 375 kilobase region analyzed. EP300WT NA-ATLL cells showed some replication pausing, likely due to some endogenous stress (Fig. 2B, D). In contrast, EP300Mut ATLL cells revealed significant replication perturbation characterized by a striking increase in dormant origin activation (Fig. 2C, E-red box; Fig. 2G-orange bars) and replication pausing (white rectangles; Fig. 2C, E). Importantly, JQAD1-mediated degradation of EP300 in EP300WT cells resulted in a striking increase in dormant origins similar to EP300Mut cells (Fig. 2F; Fig. 2G-orange lined blue bar), as compared to cells exposed to the negative control (S, S) stereoisomer (Supplementary Fig. 2B). These results suggest that lack of EP300 expression results in spontaneous replication stress that dysregulates DNA replication dynamics at fragile sites.

EP300 deficiency results in genome-wide replication stalling. To understand whether spontaneous replication stress observed in the absence of EP300 impacts replication dynamics genome-wide, we measured replication fork stalling (Fig. 3A) in EP300Mut/WT cells using DNA fiber analysis[37]. Results from this analysis showed that HU treatment resulted in replication fork stalling in all the cell lines analyzed. However, a comparison of the extent of stalling between the treated pairs revealed significantly decreased CldU: IdU ratios in EP300Mut cells (orange dots) as compared to EP300WT cells (Fig. 3B; blue dots). This was confirmed in J-ATLL cells (Supplemental Fig. 3A; J2WT and J1Mut), JQAD1-treated EP300WT NA-ATLL cells (Fig. 3B;

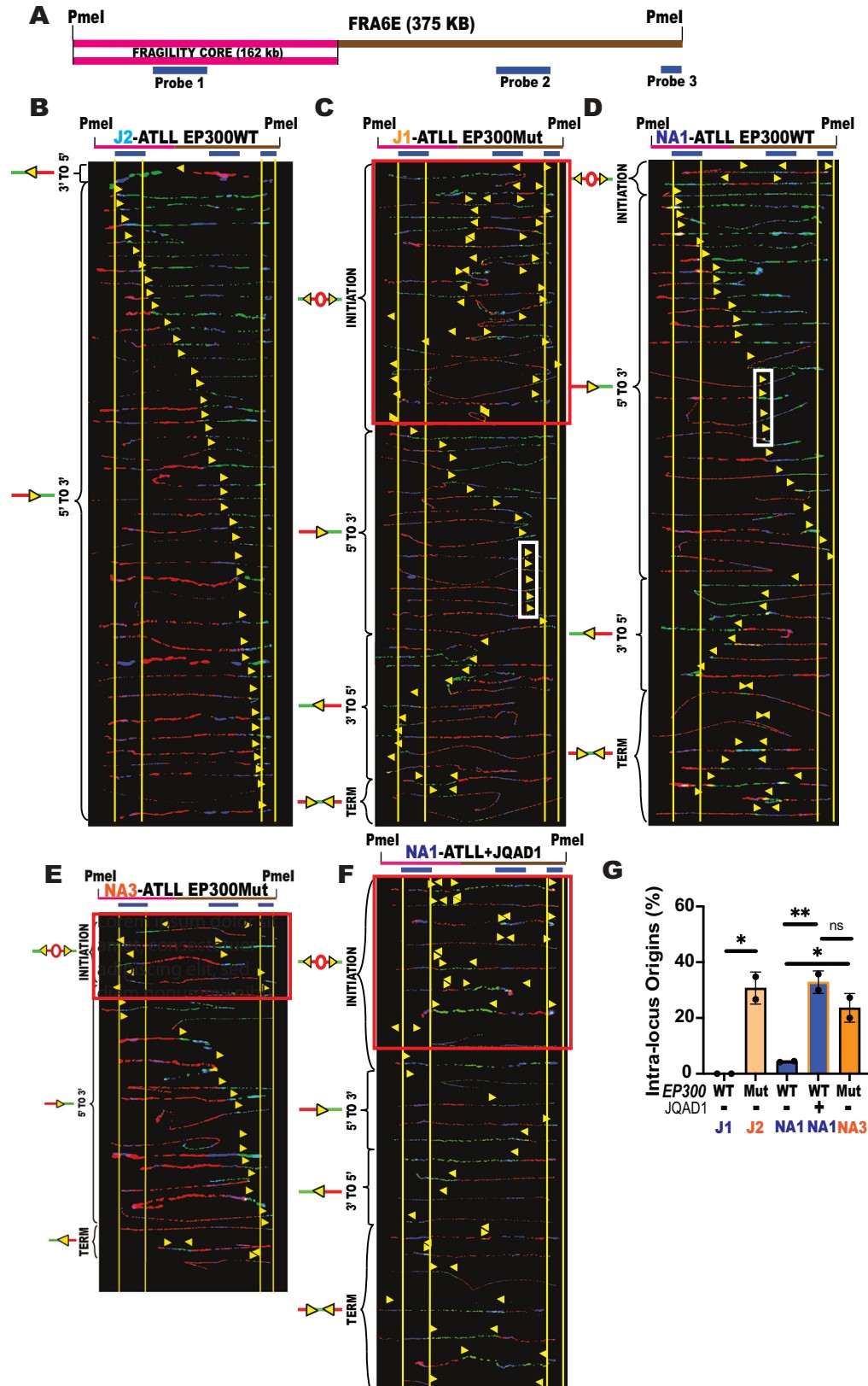

orange circles), and a non-ATLL B-lymphocyte cell line (Supplemental Fig. 3B). In contrast, EP300WT cells from each cohort did not differ significantly in CldU tract lengths, further highlighting the EP300 specificity of this phenotype.

Cells employ a specific mechanism, called fork regression, to overcome fork stalling/pausing to ultimately prevent fork collapse,

where the replication fork loses its ability to synthesize DNA[38-40] (Fig. 3A). Failure to restart the reversed forks results in nucleolytic degradation of nascent DNA[41]. To assess nucleolytic degradation, cells were first pulse labelled with IdU for 30 min followed by CldU for 30 mins and then treated with HU for 4 hours in the absence of analogs. The results revealed no significant nascent strand degradation at

**Fig. 2 | EP300-mutated cells have aberrant replication dynamics associated with increased origin firing. A** Locus map of a 375 kb region in the CFS-FRA6E obtained by *PmeI* digestion. The region includes the fragility core of CFS-FRA6E (pink line – 162 kb). The FISH probes that identify the segment are labeled in blue. Top: Locus map of the *PmeI*-digested FRA6E segment. Bottom: Aligned photomicrograph images of labeled DNA molecules from the DNA Replication program at CFS-FRA6E in (**B**) J-ATLL EP300WT (J2); (**C**) J-ATLL EP300Mut cells (J1); (**D**) NA-ATLL EP300WT cells (NA1); (**E**) NA-ATLL EP300Mut (NA2) cells; (**F**) NA-ATLL EP300WT treated with 500 nM JQAD1 for 48 h. The yellow arrows indicate the sites

along the molecules where the IdU transitioned to CldU. The molecules are arranged in the following order: molecules with initiation events, molecules with 3' to 5' travelling forks, molecules with 5' to 3' travelling forks, and molecules with termination events. **G** Percentage of molecules with initiation sites in FRA6E in J-ATLL EP300WT, J-ATLL EP300Mut, NA-ATLL EP300WT, NA-ATLL EP300Mut and NA-ATLL EP300WT cells + 500nMJQAD1 (48 h), *N* = 2. Data are presented as mean values +/− SD. *P*-values were determined by a two-tailed, Student T-test. The *p*-values are indicated as follows: * <0.03, ** <0.0021. Source data are provided as a Source Data file.

stalled forks in EP300WT cells (Fig. 3C; blue dots). However, EP300Mut cells displayed a significant increase in nucleolytic degradation, (Fig. 3C; orange dots), likely indicating a fork restart defect in the absence of EP300. JQAD1-mediated EP300 degradation in NA-ATLL EP300WT cells (NA1 and NA4) caused a similar nucleolytic degradation phenotype (Supplementary Fig. 3C), suggesting that this phenotype was linked to EP300 deficiency.

Fork restart defects in the absence of EP300 result in ssDNA gap accumulation. Fork stalling and nucleolytic degradation can lead to extensive ssDNA breaks/gaps in the genome. To assess ssDNA accumulation in the absence of EP300, we assessed BrdU incorporation under non-denaturing conditions by IF staining. The results showed a striking increase in the number of ssDNA foci detected per cell in EP300Mut NA-ATLL cells (Fig. 3D; orange dots), as compared to HU-treated EP300WT cells (blue dots). Inhibitor treatment in EP300Mut cells exacerbated the ssDNA accumulation even further. Importantly, JQAD1-mediated EP300 degradation in EP300WT cells resulted in a significant increase in spontaneous ssDNA gaps, clearly establishing that ssDNA gaps were being generated due to the absence of EP300 (Fig. 3D; JQAD1-orange circles). To establish that the source of the ssDNA were indeed collapsed forks, we co-stained for pRPASer4/8. Similar to the ssDNA data, these results revealed a striking increase in the number of pRPA4/8 foci detected per cell in EP300Mut NA-ATLL cells (orange dots), compared to EP300WT cell line (Fig. 3E; blue dots). Similar results were obtained upon EP300 degradation in a EP300WT cell line (Fig. 3E; JQAD1-orange circles). These results collectively establish collapsed forks as the primary sites of ssDNA gap formation upon EP300 loss.

Nuclease inhibition prevents ssDNA gap formation in EP300-mutated cells. The most common nucleases implicated in nucleolytic processing of nascent DNA are Mre11, DNA2, and Exo1[42]. To determine the mechanistic basis for the extensive ssDNA gaps in EP300Mut cells, we treated cells with Mirin, a potent inhibitor of Mre11 activity, in the presence or absence of HU. The immunoblotting analysis of whole cell lysates showed that the hyperphosphorylation of RPA observed largely in the EP300Mut NA-ATLL cells is rescued by Mre11 inhibition (Fig. 3F; orange bar; Supplementary Fig. 3D). Mirin treatment of EP300WT cells did not result in any significant change (Fig. 4A, B; blue bar; Supplementary Fig. 3D). Importantly, IF staining for BrdU under non-denaturing conditions after Mirin treatment resulted in a significant reduction in ssDNA gap formation in EP300Mut cells (orange dots) as compared to EP300WT cells (Fig. 3G; blue dots). Similar results were obtained upon inhibition of the DNA2 nuclease (Fig. 3H). These results clearly establish that the formation of ssDNA gaps in the absence of EP300 is driven by nuclease hyperactivity, likely due to a prominent defect in downstream fork restart machinery.

EP300-deficient cells have a prominent defect in fork restart machinery. Extensive nucleolytic processing of stalled forks has been previously described in the absence of downstream fork restart machinery (BRCA1-BRCA2-PALB2-RAD51)[41]. Importantly, EP300 has been implicated in regulating BRCA1 and RAD51 expression to promote homologous recombination[25,26]. To address this, we carried out immunoblotting analysis to measure BRCA1 and RAD51 protein expression from whole cell extracts (WCE) in EP300Mut/WT cells. The results showed no major differences in RAD51 or BRCA1 protein

expression levels irrespective of EP300 expression (Supplemental Fig. 4A). In addition, JQAD1-mediated degradation of EP300 in EP300WT cells did not alter BRCA1 expression (Supplementary Fig. 4B). Since fork degradation has been extensively associated with BRCA2 deficiency[43,44], we next evaluated BRCA2 protein expression by immunoblotting of WCE from EP300WT/Mut cells. The results revealed a prominent downregulation in BRCA2 expression in EP300Mut cells, as compared to wildtype cells (Fig. 4A, D). Importantly, while JQAD1-mediated EP300 degradation in EP300WT cells resulted in the complete loss of BRCA2 expression by day 7 of treatment, there was no change observed in EP300Mut cells (Fig. 4B–D). Next, to evaluate whether residual BRCA2 protein detected in EP300Mut cells is efficiently recruited to sites of replication stalling in response to replication stress, we carried out immunofluorescence staining to assess BRCA2 nuclear foci. Despite EP300Mut cells having extensive DNA damage and replication fork stalling (Figs. 1D, 3B), BRCA2 foci were significantly decreased in APH treated EP300Mut cells as compared to EP300WT cells (Fig. 4E). In contrast, similar levels of BRCA1 foci per cell were detected in both EP300WT and EP300Mut cells (Supplementary Fig. 4C), indicating a specific defect in BRCA2 protein expression and recruitment. BRCA2 protein is a critical mediator of the RPA-RAD51 exchange on single-stranded DNA[41,45]. To functionally validate a defect in BRCA2 protein, we next evaluated RAD51 nucleofilament formation by immunofluorescence staining. The results showed that while RAD51 foci per cell were increased in response to APH in EP300Mut cells, it was significantly lower than the RAD51 foci measured per EP300WT cell (Fig. 4F). JQAD1 mediated degradation of EP300 in wildtype cells caused a significant decrease in the observed RAD51 foci further confirming that the observed defect in RAD51 loading is EP300 and BRCA2 related (Fig. 4F). Since EP300 is a master transcriptional regulator, we next asked whether fork protection defect and the decreased BRCA2 protein expression was due to transcriptional downregulation of the proteins involved in replication fork protection by EP300. The results showed that while the transcription of a subset of genes involved in replication fork protection (e.g., EXO1, ZRANB3, FANCD2) were significantly upregulated, the majority of genes involved in this mechanism remained either unchanged or moderately upregulated (Fig. 4G). Importantly, there was no detectable decrease in the transcript levels of the genes involved in replication fork protection, including BRCA2. This was confirmed by a gene ontology analysis of genes whose expression remained relatively unchanged by EP300's absence, where most of these genes were involved in DNA repair, cell cycle, homologous recombination (fork protection) and the mitotic cell cycle (Supplementary Fig. 4D). These results collectively indicate that a downregulation of BRCA2 protein expression in EP300-deficient cells might be the underlying mechanism for defective replication fork stability and integrity. Importantly, BRCA2 downregulation appears to occur at the post-transcriptional level, indicating that EP300 is not directly controlling *BRCA2* gene transcription.

Error-prone repair mechanisms are upregulated in EP300-mutated NA-ATLL cells. Replication fork breaks in cells deficient for fork protection and HR are repaired by the microhomology-mediated break-induced replication (MMBIR) repair pathway[46,47]. To assess dependency on error-prone versus error-free replication restart in

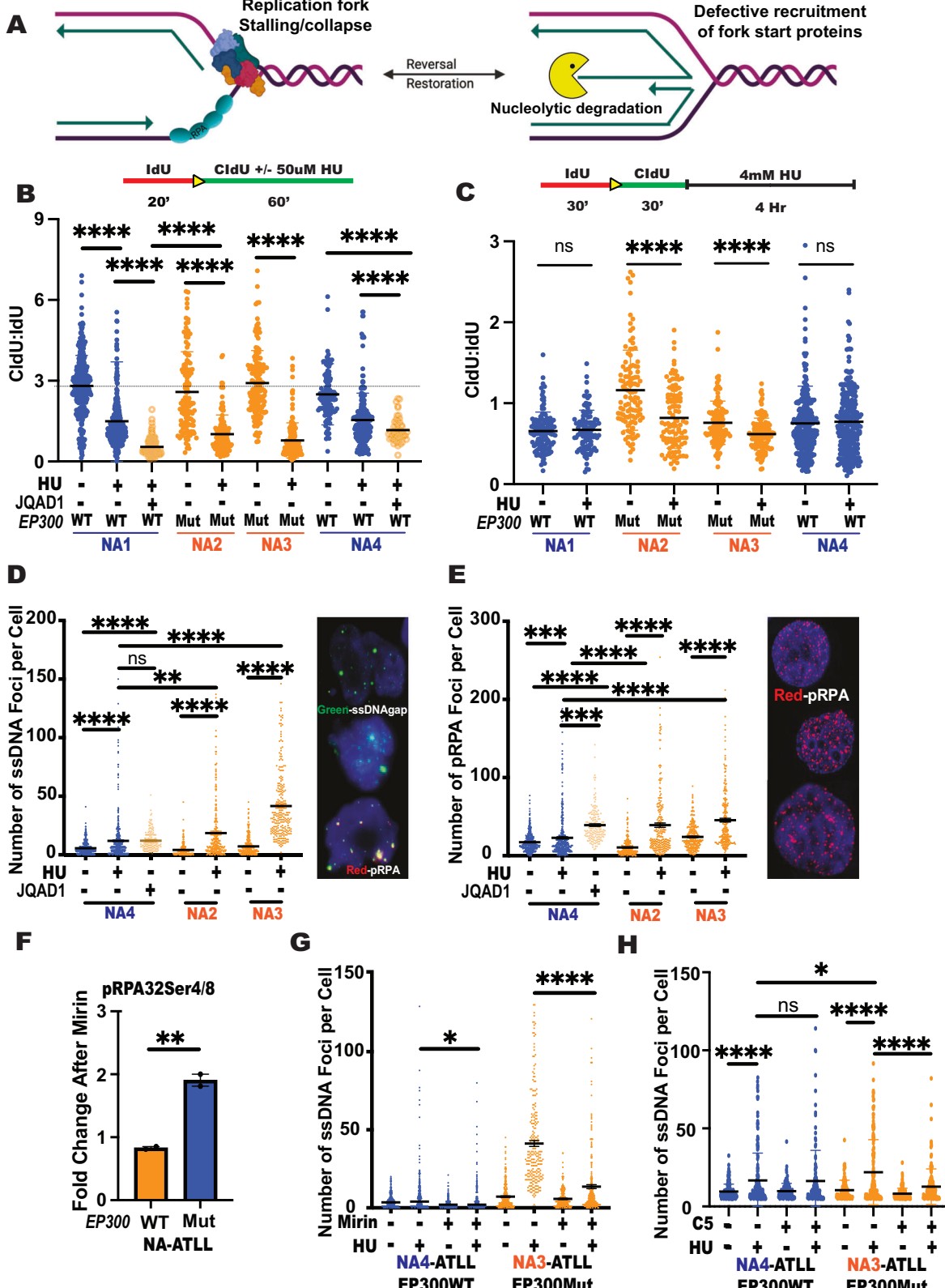

EP300Mut cells, we analyzed recruitment of PALB2, an essential factor for fork restart by HR, and POLD3, which is an essential factor of the MMBIR repair pathway[48,49]. Assessment of spontaneous PALB2 foci formation revealed a significant increase in EP300Mut cells as compared to EP300WT cells (Supplementary Fig. 4E–H). Upon HU treatment, all cell lines showed an increase in PALB2 foci formation irrespective of EP300 expression (Supplementary Fig. 4E, F).

Interestingly, treatment with an alternate replicative inhibitor, aphidicolin (DNA polymerase inhibitor) in low doses, resulted in a significant decrease in PALB2 chromatin recruitment in EP300Mut ATLL cells, compared to their untreated pairs (Supplementary Fig. 4G, H). Analysis of POLD3 chromatin recruitment to assess dependency of EP300Mut ATLL cells on error-prone fork restart revealed a significant increase in POLD3 foci per cell in EP300Mut J-ATLL and NA-ATLL cell

**Fig. 3 | EP300 deficiency results in genome-wide replication stalling and fork collapse, leading to the accumulation of extensive ssDNA gaps. A** Schematic model depicting replication fork stalling, fork protection, and nucleolytic degradation, (Created in BioRender. Madireddy, A. (2025), is licensed under CC BY 4.0, https://BioRender.com/xtm4wme). **B** DNA fiber analysis of 50uM hydroxyurea (HU) (60 min) treated EP300WT (NA1, NA4), EP300Mut (NA2, NA3) and EP300WT NA-ATLL (NA1, NA4) cells treated with negative control (S, S) stereoisomer or with JQAD1 (500 nM for 48 h) treatment to assess replication fork stalling. The fork rate (CldU/IdU ratio) is indicated. $N = 3$. **C** DNA fiber analysis measuring nucleolytic degradation after 4 mM HU treatment in EP300WT, EP300Mut NA-ATLL cells. The fork rate (CldU/IdU ratio) is indicated, $N = 3$. **D** Analysis of the number of single-stranded gaps (ssDNA)/Iododeoxyuridine (IdU) foci (green) per cell nuclei (DAPI, blue) in EP300WT (NA4) and EP300Mut (NA2, NA3) cells exposed to 2 mM HU (4 h), and EP300WT (NA4) cells treated with JQAD1 (500 nM). ssDNA gaps are measured by IdU incorporation under non-denaturing conditions. Representative images are shown on the right, $N = 3$. **E** Analysis of number of pRPA Ser4/8 foci (red) per cell nuclei (DAPI, blue) in EP300WT (NA4) and EP300Mut (NA2, NA3) cells exposed to 2 mM HU (4 h), and EP300WT (NA4) cells treated with 500 nM JQAD1. Representative images are shown on the right, $N = 3$. **F** Protein expression of phospho-RPA Ser4/8 in the presence or absence of EP300, presented as a fold change quantified from the immunoblot in Supp. Fig. S3D. **G** Analysis of the number of single-stranded gaps/breaks (ssDNA)/Iododeoxyuridine (IdU) foci per cell nucleus (DAPI, blue) in EP300WT (NA4) and EP300Mut (NA3) cells exposed to 2 mM HU (4 h), in the presence or absence of 50uM Mirin (4 h) (Mre11 nuclease inhibitor), $N = 3$. **H** Analysis of the number of single-stranded gaps/breaks (ssDNA)/ Iodo-deoxyuridine (IdU) foci per cell nucleus (DAPI, blue) in EP300WT (NA4) and EP300Mut (NA3) cells exposed to 2 mM HU (4 h), in the presence or absence of 20uM C5 (24 h) (DNA2 nuclease inhibitor), $N = 3$. For all experiments, data are presented as mean values +/− SD. *P*-values were determined by a two-tailed Student T-test. *P*-values are indicated as follows: * <0.03, ** <0.0021, *** <0.0002, **** <0.0001. Scale bar 10 μm. N represents three experimental replicates from independent cultures of cells. Source data are provided as a Source Data file.

lines, as compared to EP300WT cells (Fig. 4H; Supplementary Fig. 4I). Importantly, Mre11 inhibition led to a significant decrease in the POLD3 foci per cell EP300Mut NA-ATLL cells (Supplementary Fig. 4J), showing that error-prone repair was the prevalent mechanism of replication-associated repair of collapsed forks in these cells.

Similar to BRCA-deficient cells, EP300-deficient cells are sensitive to inhibition of PARP, POLQ, and Rev1. It has been shown that BRCA-deficient cells are sensitive to PARP (Poly (ADP-ribose) polymerase) inhibitors due to synthetic lethality[50]. More recently, second-generation compounds that can target HR-deficient cells while overcoming PARPi resistance have been developed. These involve two major translesion synthesis polymerases that mediate DNA damage tolerance mechanisms that can maintain the viability of HR or BRCA-deficient cells. The first is REV1-Polζ[51,52] and the second is Polymerase theta (POLθ) mediated end joining (TMEJ), both of which mediate mutagenic repair of ssDNA gaps[53–55], that are highly elevated in EP300Mut cells (Fig. 3D). To evaluate whether EP300Mut cells are sensitive to PARPi, Rev1i and POLQi, we exposed EP300Mut/WT NA-ATLL cells to these inhibitors and assessed apoptotic fractions in the cells by flow cytometry, co-staining for Annexin V (apoptotic marker) and PI. These results showed a significant increase in the percentage of cells undergoing apoptosis (early and late) in EP300Mut cells as compared to EP00WT cells (Fig. 4I–K), in response to each of the inhibitors individually. Interestingly, EP300Mut cells were more sensitive to Rev1i and POLQi as compared to PARPi. Importantly, combining POLθi + PARPi and Rev1i+PARPi treatments, augmented the apoptotic fractions in EP300Mut NA-ATLL cells, as compared to EP300WT cells (Fig. 4I–K). Since PARPi is currently in clinical use for HR deficient cancers, we next assessed in vitro toxicity to PARPi in EP300Mut/WT NA-ATLL cell lines which revealed that EP300Mut cells are quite sensitive to Olaparib treatment, as compared to the EP300WT NA-ATLL cells (Supplementary Fig. 4K). The Olaparib IC50 range for the NA-ATLL (20−39.5 uM) is very comparable to that reported for Olaparib-sensitive breast cancer cell lines (10−60 uM) based on a 3-day cell viability assay[56]. These results suggest that EP300 deficiency and PARPi are synthetic lethal in ATLL. In addition, these results highlight the potential efficacy of PARP, POLθ, and REV1 inhibitors in limiting cell viability in EP300Mut cells.

EP300-deficient NA-ATLL cells have under-replicated DNA and persistent genomic instability that results in micronuclei formation and inherited DNA lesions in G1. Next, we wanted to determine whether the defective replication-associated ssDNA gaps and the DNA damage observed in EP300Mut S-phase cells persisted upon transition of cells into G2/M. Perturbed DNA replication has been associated with incompletely replicated DNA, visualized as ultra-fine DNA bridges (UFB) during mitosis. UFBs that originate from CFS loci can be identified by twin foci formed by the Fanconi anemia complementation group D2/I (FANCD2/FANCI) proteins at the termini of the UFB on each

chromatid[57]. To determine the presence of under-replicated DNA in NA-ATLL cells, we measured the persistence of FANCD2 twin foci in mitotic cells. This analysis revealed that EP300Mut NA-ATLL cells had a striking spontaneous increase in FANCD2 twin foci in mitotic cells, indicating incompletely replicated DNA arising from CFS loci as the primary source of post-replicative distress in the absence of EP300 (Fig. 5A). Replication inhibitor treatment further exacerbated the number of FANCD2 twin foci in NA-ATLL mitotic cells in a EP300-dependant manner. The condensation of incompletely replicated regions triggers the completion of DNA replication during G2/M by the process of Mitotic DNA Synthesis (MiDAS), predominantly at CFS[58,59]. Analysis of MiDAS in NA-ATLL cells revealed that while EdU synthesis between FANCD2 twin-foci was present in all NA-ATLL cells, there was no significant increase in EdU after replicative inhibition in EP300Mut cells (Fig. 5B). Importantly, while JQAD1-mediated EP300 degradation in EP300WT cells resulted in a spontaneous increase in FANCD2 twin foci, there was no significant increase in EdU-MiDAS in these cell lines (Supplementary Fig. 5A, B). These results suggest a possible defect in MiDAS in the absence of EP300.

Under replicated DNA and lagging chromosomes leads to micronuclei formation in the subsequent cell cycle. Micronuclei (MN) are aberrant nuclear structures that transiently encapsulate mis-segregated DNA outside of the nucleus[60]. To address this, we next measured the percentage of cells with micronuclei positively stained for pRPA and FANCD2. Both analyses revealed that the micronuclei observed in NA-ATLL EP300Mut cells have a significant increase in both pRPA staining and FANCD2 foci (Fig. 5C, D), clearly indicating replicative defects as the source of micronuclei formation in the absence of EP300. Given this elevated genomic instability in EP300Mut mitotic cells, we assessed mitotic mis-segregation defects in the subsequent cell cycle. Persistent unresolved DNA damage in newly formed daughter cells is localized to 53BP1 nuclear bodies in G1[61,62]. Quantification of 53BP1 nuclear bodies by immunofluorescence staining in cyclin A negative G1 cells revealed a significant increase in 53BP1 nuclear bodies in the absence of EP300 (Fig. 5E). These results collectively showed that unresolved DNA damage in NA-ATLL cells is transmitted as inherited lesions in the subsequent cell cycle.

EP300 mutated cancers closely resemble BRCA-deficient tumors. In normal cells, replication fork stalling in response to replication inhibition results in fork reversal at nascently synthesized DNA followed by RAD51 loading and the efficient restart of stalled forks in the presence of downstream effectors such as BRCA1/BRCA2/ PALB2, thus maintaining genome stability (Fig. 6, top). However, in the absence of the EP300, elevated endogenous replication stress results in dormant origin firing at common fragile sites, and increased genome-wide replication pausing. Defective downstream effectors, such as BRCA2, lead to excessive nucleolytic degradation at regressed forks resulting in increased ssDNA at collapsed forks. This in turn instigates POLD3-

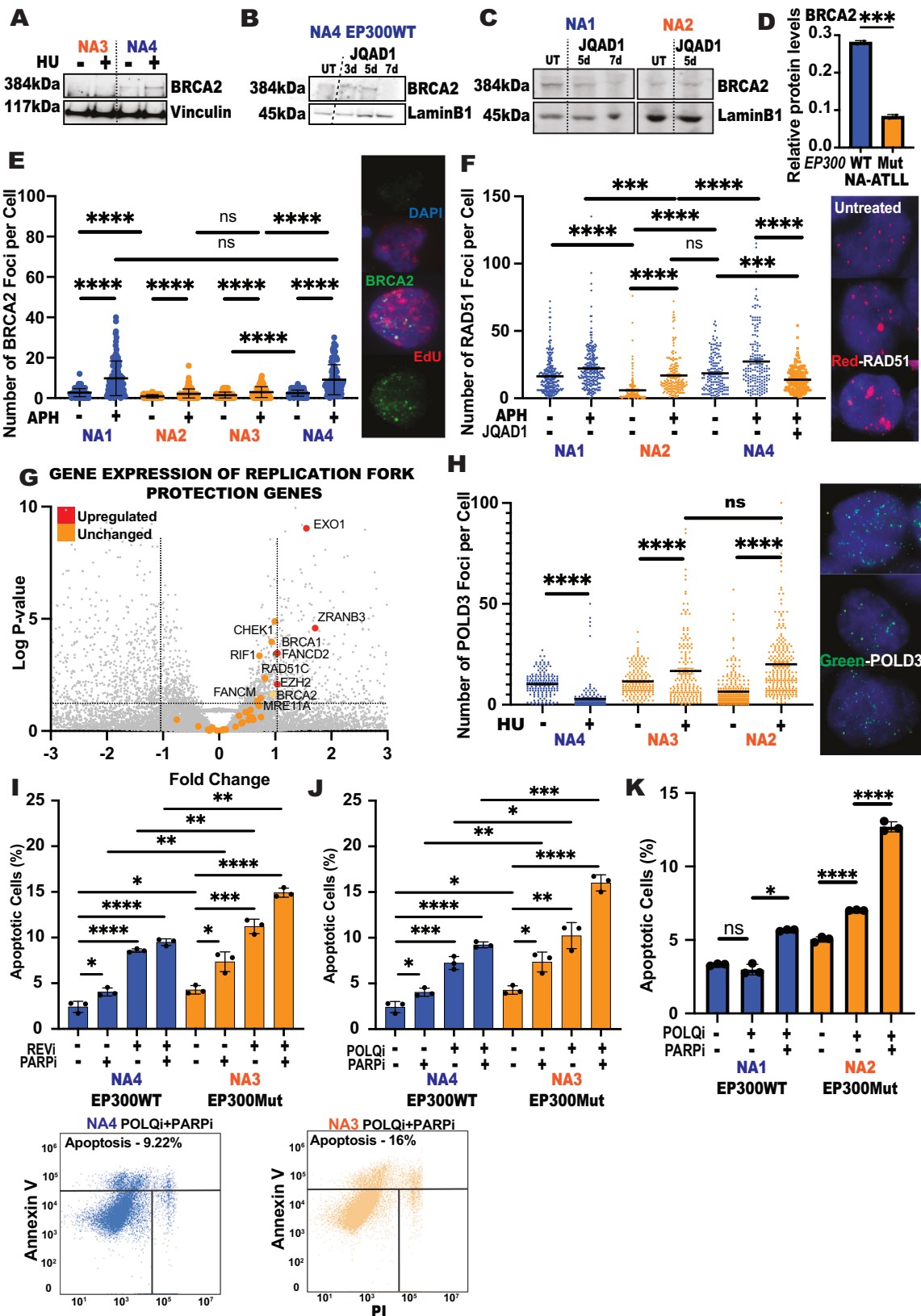

mediated replication restart during the S-phase, collectively mimicking phenotypes observed in BRCA-deficient cancers. Upon transitioning to G2/M, EP300-deficient cells display ineffective MiDAS, persistent under-replicated DNA and increased miss-segregation of DNA during mitosis to daughter cells in G1 (Fig. 6, bottom). Taken together, the results from this study indicate that the EP300 protein plays an important previously undiscovered role in replication fork protection, which is essential to maintaining genome stability.

## Discussion

In this study, we reveal a novel role for EP300 in facilitating DNA replication upon replication stress. Replication stress refers to changes

**Fig. 4 | EP300-deficient cells have a prominent defect in downstream fork restart machinery. A** Expression levels of BRCA2 protein from WCE in EP300WT and EP300Mut NA-ATLL cells treated with 2 mM HU (4 h), by immunoblotting. Expression levels of Vinculin were used as a loading control. **B** Expression levels of BRCA2 protein in the NA4 EP300WT NA-ATLL cell lines treated with negative control (S, S) stereoisomer compared to cells 3,5,7 days after 500 nM JQAD1 treatment (48 h). Expression levels of LaminB1 were used as a loading control. **C** Expression levels of BRCA2 protein in the NA1 EP300WT NA-ATLL cell line with negative control (S, S) stereoisomer compared to cells 5,7 days after 500 nM JQAD1 treatment (48 h) and NA2 EP300Mut cell line 5 days after JQAD1 treatment. Expression levels of LaminB1 were used as a loading control. **D** Relative expression of BRCA2 protein between EP300WT and EP300Mut cells. **E** Analysis of the number of BRCA2 foci (green) per EDU-positive (red) cell nucleus (DAPI, blue) in EP300WT (NA1, NA4), and EP300Mut (NA2, NA3) cells exposed to 0.4 uM APH (O/N). Representative images are shown on the right, *N* = 3. **F** Analysis of the number of RAD51 foci (red) per cell nucleus (DAPI, blue) in EP300WT (NA1, NA4) +/− 0.4 uM APH (O/N), EP300Mut +/− 0.4 uM APH (O/N) and EP300WT (NA4) treated with 500 nM JQAD1 (48 h). Representative images are shown on the right, *N* = 3.

**G** Volcano plot comparing gene (mRNA) expression levels, quantified and presented as a fold change in EP300WT (NA1, NA4) and EP300Mut (NA2, NA3) cell lines after RNA-seq analysis. **H** Analysis of the number of POLD3 foci (green) per cell nucleus (DAPI, blue) in EP300WT (NA4) and EP300Mut (NA2, NA3) cells exposed to 2 mM HU (4 h). Representative images are shown on the right, *N* = 3. **I** Quantification of the percentage of AnnexinV/PI +ve cells assessed by flow cytometry in EP300Mut, EP300WT NA-ATLL cells +/− REV1i (10uM JHRE06 for 24 h) and +/− PARPi (10uM Olaparib for 72 h), *N* = 3. **J** Quantification of percentage of AnnexinV/PI +ve cells assessed by flow cytometry in EP300Mut, EP300WT NA-ATLL cells +/− POLQi (20uM ART558 for 72 h) and +/− PARPi (10uM Olaparib for 72 h). Representative images are shown on the right, *N* = 3. **K** Quantification of the percentage of AnnexinV/PI +ve cells assessed by flow cytometry in EP300Mut (NA2), EP300WT (NA1) cells +/− POLQi (20uM ART558 for 72 h) and +/− PARPi (10 uM Olaparib for 72 h). For all experiments, data are presented as mean values +/- SEM. *P*-values were determined by a two-tailed Student T-test. The *p*-values are indicated as follows: * <0.03, ** <0.0021, *** <0.0002, **** <0.0001. Scale bar 10 μm. N represents experimental replicates from independent cultures of cells. Source data are provided as a Source Data file.

in the cellular microenvironment that results in increased replication pausing, increased replication origin firing and variations in replication fork speeds. Here, we show that EP300 deficiency invokes a spontaneous increase in replication stress. Through this study, we highlight a role for the EP300 protein in mediating replication fork protection at sites of replication stalling. In EP300's absence, defective replication fork protection is linked to a significant decrease in the BRCA2 protein expression, resulting in the accumulation of single-stranded gaps. While EP300 is a well-established transcriptional co-activator, our study suggests that decreased BRCA2 protein expression is not transcriptionally altered by EP300 loss. On the contrary, we believe that the EP300-associated decrease in BRCA2 is controlled post-transcriptionally by a currently unknown mechanism. Notably, our investigations pinpoint the S-phase as a targetable vulnerability in EP300Mut NA-ATLL, offering a promising avenue for intervention. In addition, our results from this study draw strong parallels between EP300-mutated cells and BRCA-deficient cells. This is not only highlighted by the enhanced sensitivity of EP300-mutated cells to PARP inhibition but also the acute sensitivity of EP300Mut cells to Rev1 and Pol Theta inhibition, the key DNA damage tolerance mechanism(s) that BRCA-deficient cells rely on. Overall, these findings pave the way towards evaluating novel therapeutic strategies for EP300-mutated cancers such as ATLL.

Despite decades of research, there is no clear understanding of the independent roles of EP300 and CBP in human cells. This is largely due to the high degree of sequence homology between EP300 and CBP in multiple homologous domains, which has resulted in most functional studies in the literature examining the loss of EP300/CBP as a single unit[63–69]. However, despite their similarities in structure, EP300 and CBP appear to have non-overlapping cellular functions, distinct binding partners, and chromatin recruitment sites. The precise and time-resolved roles of EP300 in different biological processes has been historically understudied due to a lack of inhibitors specific to EP300. Current commercially available inhibitors to the HAT and Bromodomains of EP300 have nonselective activity against CBP as well. Despite this, there is an unmet need to identify signaling cascades and cellular pathways dysregulated in the absence of EP300 to understand the distinct functions of each protein in human cell. In our studies, we have utilized a unique collection of ethnically diverse patient-derived EP300-mutated cell lines (NA-ATLL and J-ATLL) and a novel EP300-specific PROTAC degrader, JQAD1[13], to address this major gap in understanding the functional consequences of EP300 loss.

The spectrum of mutations observed in EP300Mut NA-ATLL cells include missense, splice site, and truncating mutations (Supplementary Table 1). While the EP300Mut NA-ATLL cells we have analyzed in the paper show very low to undetectable EP300 protein expression,

wildtype cells show prominent EP300 signal by immunoblotting (Supplementary Fig. 1A)[6,70]. In addition, we find that the CBP protein is expressed in all EP300Mut/WT NA-ATLL cells analyzed (Supplementary Fig. 1A). This suggests that the mutations are likely loss of function *EP300* mutations rendering the protein inactive, the mechanism of which is unclear. To address inter-patient heterogeneity between cell lines, paired samples were treated with either the EP300-specific PROTAC degrader, JQAD1 or its negative control (S,S) stereoisomer. While a number of experimental endpoints were achievable 24–48 h after JQAD1 treatment, some endpoints such as H3K27Ac changes, replication abnormalities and the BRCA2 protein dysregulation took anywhere from 3 to 7 days to manifest. This delayed phenotype might be explained by the fact that chromatin bound EP300 is harder to access. JQAD1 is built using an E3 receptor recruiting ligand to the E3 receptor CRBN, which is predominantly found in the cytoplasm, rather than the chromatin compartment where EP300 is engaged. Further, the downstream consequences of epigenetic alterations take time to translate into observable cellular phenotypes. This has been previously demonstrated for compounds targeting epigenetic regulators such as EZH2[71] or DNMT1[72]. While it is true that cell line models have limitations in fully representing the diverse complexities of EP300 mutations seen in clinical settings, they provide an excellent avenue to explore and unveil new biological pathways and mechanisms. Future experiments utilizing patient derived xenograft models will be invaluable in establishing the clinical relevance of the mechanisms uncovered and testing of potential therapeutic strategies uncovered by this study.

BRCA2 is a critical tumor suppressor involved in maintaining genomic integrity, notably through homologous recombination (HR) and the protection of stalled replication forks. When replication forks stall due to DNA damage or other impediments, BRCA2 stabilizes them, preventing their degradation by nucleases like MRE11[44]. BRCA2 protein expression has been shown to increase with cell cycle progression of cells into the S-phase and further increases upon replication stress[73,74]. When cells are resting in G0/G1, BRCA2 expression is expected to be low. We find that BRCA2 protein loss in EP300Mut cells is not associated with a G0/G1 arrest (Supplementary Fig. 1I). On the contrary, EP300Mut cells display a prominent S-phase arrest. While one would expect to see an increase in BRCA2 protein expression, S-phase arrest in EP300Mut cells is accompanied by a prominent decrease of BRCA2 protein levels clearly demonstrating that BRCA2 protein expression changes are not attributable to changes in the cell cycle. An important question raised by our study is the potential mechanistic link between EP300 depletion and BRCA2 downregulation. Our results reveal that *BRCA2* gene expression remains unchanged by EP300 loss. This extends to the transcriptional program of genes involved in several DNA repair mechanisms including

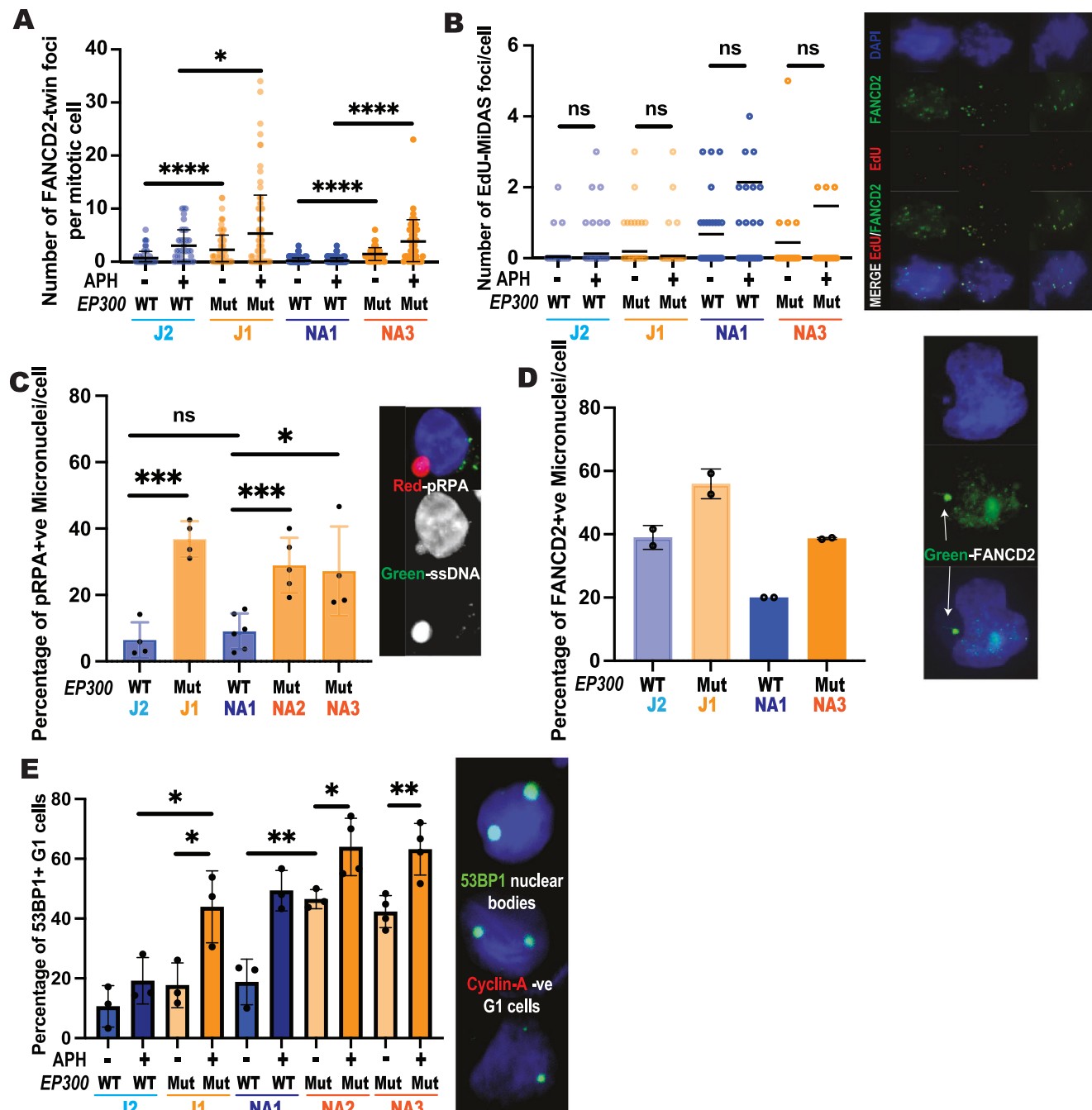

**Fig. 5 | EP300 mutated cells have under-replicated DNA that results in inherited DNA lesions in G1. A** Analysis of number of FANCD2 twin-foci (green) per mitotic cell nucleus (DAPI, blue) in EP300WT and EP300Mut cells exposed to 0.4uM Aphidicolin (APH) overnight (O/N), representative images are shown on the right hand most corner, $N = 2$. **B** Analysis of number of EdU foci (red), sandwiched between FANCD2 twin-foci, per mitotic cell nucleus (DAPI, blue) in EP300WT and EP300Mut cells exposed to 0.4 uM APH overnight (O/N), representative images are shown on the right, $N = 2$. **C** Analysis of the percentage of pRPA (red) positive micronuclei in EP300Wt and EP300Mut cells, in the absence of any inhibitor treatments. Representative images are on the right, for samples J1, J2, NA1, and NA3, $N = 4$, and for NA2, $N = 5$. **D** Percentage of FANCD2 (green) positive micronuclei in EP300WT and EP300Mut cells, in the absence of any inhibitor treatments. Representative images are on the right, $N = 2$. **E** Analysis of the number of 53BP1 nuclear bodies positive in G1 cells (cyclin A negative) in EP300WT and EP300Mut cells exposed to 0.4uM APH overnight (O/N), representative images are shown on the right, For J1, J2, NA1, and NA2 (UT) $N = 3$ and for NA2 (APH) and NA3, $N = 4$. For all experiments, data are presented as mean values +/- SD. *P*-values were determined by a two-tailed Student's T-test. The *p*-values are indicated as follows: * <0.03, ** <0.0021, *** <0.0002, **** <0.0001. Scale bar 10 μm. N represents two/three experimental replicates from independent cultures of cells. Source data are provided as a Source Data file.

homologous recombination repair in EP300Mut NA-ATLL cells. This data suggests that the variation in BRCA2 protein expression is likely post-transcriptionally controlled. Posttranscriptional control involves mechanisms such as translational efficiency, post-translational modification, and protein stability. Further investigation into these mechanisms will be critical in understanding the differences in the core replication machinery and fork associated proteome, in the absence of EP300.

There is currently a keen interest in understanding the potential mechanisms underlying EP300 mutation-related tumorigenesis and

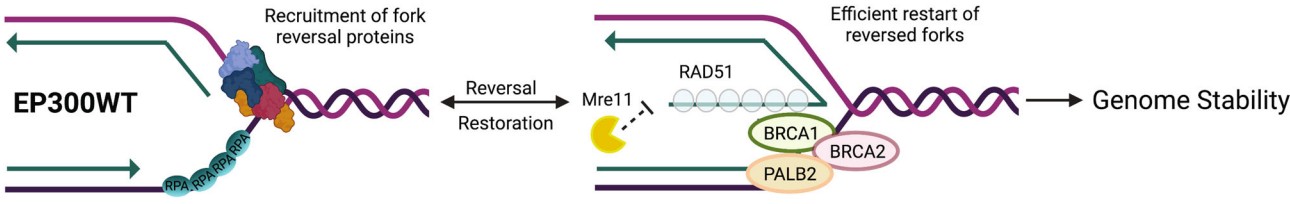

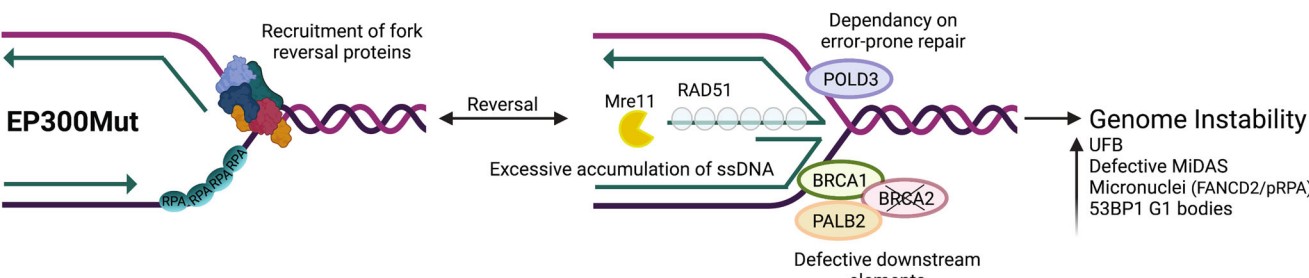

**Fig. 6 | EP300-mutated cancers closely resemble BRCA-deficient tumors.** (Top) In EP300WT cells, replication fork stalling in response to replication inhibition (depletion of nucleotide pools, DNA polymerase inhibition, oncogene activation), results in fork reversal at nascently synthesized DNA followed by RAD51 loading and the efficient restart of stalled forks in the presence of downstream effectors such as BRCA1/BRCA2/ PALB2, thus maintaining genome stability. (Bottom) However, in EP300Mut cells, elevated endogenous replication stress results in dormant origin firing at common fragile sites and increased genome-wide replication pausing. While fork regression followed by RAD51 leading occurs in these cells, defective downstream effectors, such as BRCA2, lead to excessive nucleolytic degradation at regressed forks, resulting in increased ssDNA at collapsed forks. This in turn, triggers POLD3-mediated replication restart during the S-phase, collectively mimicking phenotypes observed in BRCA-deficient cancers. Upon transitioning to G2/M, EP300-deficient cells rely on POLD3-dependent MiDAS to overcome under-replicated DNA, which is largely insufficient to repair the excessive persistent damage, resulting in miss-segregation of DNA during mitosis to daughter cells in G1. "EP300 mutated cancers closely resemble BRCA-deficient tumors,". Created in BioRender. Madireddy, A. (2025), is licensed under CC BY 4.0, https://BioRender.com/9lfufms.

their overall implications for cancer therapy. The collective dysregulation of EP300/CBP has been shown to contribute to the development and progression of various types of cancers[63]. For instance, EP300/CBP are responsible for acetylating histone proteins, which impacts chromatin structure and gene expression. Mutations or dysregulation can lead to altered histone acetylation patterns (e.g., focal depletion of H3K27ac), resulting in aberrant silencing or activation of genes essential for normal hematopoiesis, cell growth, and differentiation. In addition, this dysregulation has been shown to affect the transcription of both tumor suppressor genes or oncogenes[75], depending on the specific cancer type and the cellular context. This context-dependent behavior highlights the complexity of its role in tumorigenesis and the need for further research to fully understand its implications. Through our study, we have uncovered replication fork protection as a novel mechanism abrogated by the independent loss of EP300, likely driving aggressive tumorigenesis. Given the current lack of strategies to treat EP300-mutant cancers such as Adult T-cell Leukemia Lymphoma, this study paves the way towards uncovering new therapeutic venues based on our identification of novel molecular pathways dysregulated primarily in the absence of EP300. Our studies have revealed a selective sensitivity of EP300Mut NA-ATLL cells to PARP inhibition, Polymerase Theta inhibition, and Rev1 inhibition. These form the basis for ongoing work studying the interactions between these targets, which provide a foundation for potential new therapeutic strategies in this fatal disease.

## Methods
### Cell culture
Human cell line GM03798 (wild type), Epstein–Barr virus-transformed lymphoblasts were obtained from Coriell Cell Repositories and were grown in RPMI 1640 (Gibco,11875119) supplemented with 10% FBS and penicillin/streptomycin (Gibco,15140122). Japanese ATL43+ and ATL43- cells were cultivated in RPMI 1640 medium supplemented with 10% fetal bovine serum (FBS), 1% penicillin/streptomycin, and with or without interleukin 2 (IL-2) (200-02-100UG, Peprotech), respectively. North American ATLL cells were cultivated in IMDM medium supplemented with 20% human serum, 1% penicillin/streptomycin, and 100 unit/ml IL-2. Cells were treated with the different concentrations of the following drugs: Hydroxyurea (H8627-1G; Sigma); Aphidicolin (A0781, Sigma); Camptothecin (ab120115; Abcam). Cell lines were regularly tested for mycoplasma contamination.

### DNA fiber analysis
DNA fibers were stretched and prepared using a modification of a procedure described previously[76,77]. Briefly, cells were pulse-labeled with 30 μM IdU/CldU for 20 min each. Labelled cells were resuspended in cold PBS at $1 \times 10^6$ cells/ml. Two μl of cell suspension was spotted onto a clean glass slide and to lyse it, 10 μl of spreading buffer (0.5% SDS, 200 mM Tris-HCl (pH 7.4), and 50 mM EDTA) was added. The cells were incubated for 6 min, and the slides were tilted to spread the DNA. Slides were either fixed in methanol and acetic acid (3:1) for 2 min, followed by denaturation with 2.5 M HCl for 30 min at room temperature. Alternatively, the DNA was denatured with sodium hydroxide in ethanol and then fixed with glutaraldehyde. The slides were blocked with 1% BSA for at least 20 min. The slides were incubated with the antibodies as described above for SMARD. The coverslips were mounted with ProLong gold antifade reagent (Invitrogen, P36934) after a final PBS/CA630 rinse. Fluorescence microscopy was carried out using a Zeiss microscope to monitor the IdU/CldU nucleoside incorporation.

### Immunofluorescence
Cells were harvested and seeded on slides using Cytospin4. Cells were then treated with 0.4uM Aphidicolin overnight(16 h) or Hydroxyurea

(HU) to induce replication stress. Treatment was withdrawn and media was removed. Slides seeded with cells were fixed and permeabilized for 15 min with 0.5% Triton in DPBS containing 4% formaldehyde. The slides were then washed twice with DPBS- to be used for IF, or they can be stored at 4 °C for a maximum two weeks. Primary antibodies used were anti- Phospho-Histone H2A.X (Ser139) (Cell Signaling Technology, 9718 T or EMD Millipore, 05-636, 1:1000), antiBRCA2 (Merck Millipore OP95, 1:1000) and anti-RAD51 (CosmobioUSA BAM-70-002-EX, 1:1000), anti-POLD3 (abnova, H00010714-M01, 1:1000). Slides were then mounted with ProLong Gold anti fade reagent with DAPI (Invitrogen, P36935). Images were captured using a Zeiss Axiom fluorescence microscope.

## Western blotting

Cells were harvested after treatment by centrifugation, followed by PBS wash. Cells were lysed with 2X Laemmli buffer and lysates were denatured at 100 °C for 15 min. Proteins were separated using NuPAGE 4-14% Bis-Tris Mini Gels (Thermo Scientific, NP0321BOX) or 3–8% Tris-Acetate mini gels (Thermo Scientific, (EA0375BOX) and transferred to nitrocellulose membrane. Membranes were blocked with AdvanBlock (Advansta, R-03726-E10) blocking buffer for 1 h at room temperature and then incubated with primary antibodies overnight at 4 °C. Membranes were washed 3 times for 15 min and then incubated with horseradish peroxidase (HRP)-linked secondary antibodies for 1 h at room temperature. Proteins were then detected by chemiluminescence. Time Course Experiment: Cells were grown in corresponding media and treated with 2 mM HU for 4 h. Cells were released from HU into fresh media and collected at various time points after release: 0, 4, 8, 12, 24, 28, 32, and 48 h after release. Cells were then harvested and lysed for immunoblotting. Antibodies for western blot included anti-Vinculin (Sigma Aldrich – V9131, 1:1000), anti-pH2AX (Cell Signaling – 9718 T, 1:1000), anti-pCHK1 (s317) (Cell signaling –12302, 1:1000), anti-pRPA (ser4, ser8) (Bethyl Lab A300-245A-M, 1:1000), anti-CHK1 (Cell Signaling – 2360, 1:1000), anti-RPA (Bethyl – A300-244A, 1:1000), anti-H3 (Cell Signaling – 12648 T, 1:5000), anti-H3K27ac (Cell Signaling 8173 T, 1:1000), anti-EP300 (ABCAM- ab10485, Cell Signaling- 70088 1:1000), anti-BRCA1 (EDM Millipore – 07434), anti-BRCA2 (EDM Millipore – OP95, 1:1000), anti-rad51 (Abcam – ab63801, CosmobioUSA-Bam-70-002-EX, 1:500), phospho-ATR (Cell Signaling, Cat#30632, 1:2000), Anti-CBP (Cell Signaling Technology, Cat# 7389, 1:500).

## Single Molecule Analysis of Replicated DNA (SMARD)

SMARD analysis was carried out using a previously described procedure[34,78,79]. Briefly, exponentially growing cells were cultured in media containing 30 μM 5-iodo-2′-deoxyuridine (IdU) at 37 °C for 4 h (MP Biomedicals, 0210035701). After 4 h, the cells were centrifuged at 100–110 g for 5 min and the media containing IdU was removed. The cells were then cultured in fresh RPMI medium containing 30 μM 5-chloro-2′-deoxyuridine (CldU) (MP Biomedicals, 0210547883) and incubated for an additional 4 h. After 4 h, the cells were collected by centrifugation and were resuspended at $3 \times 10^7$ cells per ml in PBS. The cells were resuspended in an equal volume of molten 1% InCert agarose (Biorad Laboratories,1613111) in PBS. DNA gel plugs were made by pipetting the cell-agarose mixture into a chilled plastic mold with 0.5 x 0.2 cm wells with a depth of 0.9 cm. The gel plugs were allowed to solidify on ice for 30 min. The cells in the plugs were lysed in buffer containing 1% n-lauroylsarcosine (Sigma-Aldrich), 0.5 M EDTA, and 20 mg/ml proteinase K. The gel plugs were incubated at 50 °C for 3 days and treated with fresh proteinase K at 20 mg/ml concentration (Invitrogen, 25-530-015), every 24 h. The Proteinase K-digested plugs were then rinsed in Tris-EDTA (TE) and subjected to phenylmethanesulfonyl fluoride (PMSF) (Sigma-Aldrich, P7626) treatment. To prepare the cells for restriction enzyme digestion, the plugs were washed with 10 mM MgCl$_2$ and 10 mM Tris-HCl (pH 8.0), and the genomic DNA in the gel plugs was digested with 70 units of PmeI or SbfI (New England

BioLabs Inc.) at 37 °C overnight. The digested gel plugs were rinsed with TE and cast into a 0.7% SeaPlaque GTG agarose gel (Lonza Rockland, Inc., BMA50111) for size separation of DNA by pulse field gel electrophoresis. Gel slices from the appropriate positions in the pulsed-field electrophoresis gel were melted at 72 °C for 20 min. The melted agarose was digested with Beta agarase 1 enzyme (4 units per 70 μL of agarose mixture; New England Biolabs, M0392) by incubating the agarase digestion buffer (1X TE pH 8, 1% 5 M NaCl, 0.1% BME) at 45 °C for 4 h. The resulting DNA was pipetted along one side of a coverslip that had been placed on top of a 3-aminopropyltriethoxysilane (Sigma-Aldrich, A3648)-coated glass slide and allowed to enter by capillary action. The DNA was denatured with sodium hydroxide in ethanol and fixed with glutaraldehyde.

The slides containing the DNA were hybridized overnight with biotinylated probes (represented as blue bars on the CFS locus maps). The next day, the slides were rinsed in 2 × SSC (1× SSC is 0.15 M NaCl plus 0.015 M sodium citrate) 1% SDS and washed in 40% formamide solution containing 2 × SSC at 45 °C for 5 min and rinsed in 2 × SSC-0.1% IGEPAL CA-630. Following several detergent rinses (4 times in 4× SSC-0.1% IGEPAL CA-630), the slides were blocked with 1% BSA for at least 20 min and treated with Avidin Alexa Fluor 350 (Invitrogen Molecular Probes, A-11236) for 20 min. The slides were rinsed with PBS containing 0.03% IGEPAL CA-630, treated with biotinylated anti-avidin D (Vector Laboratories, BA-0300) for 20 min, and rinsed again. The slides were then treated with Avidin Alexa Fluor 350 for 20 min and rinsed again, as in the previous step. The slides were incubated with the IdU antibody, a mouse anti-iododeoxyuridine (BD Biosciences, 347580, 1:15), the antibody specific for CldU, a monoclonal rat anti-iododeoxyuridine (anti-IdU) (Abcam, ab6326, 1:15), and biotinylated anti-avidin D for 1 h. This was followed by incubation with Avidin Alexa Fluor 350 and secondary antibodies, Alexa Fluor 568 goat anti-mouse IgG (H + L) (Invitrogen Molecular Probes, A-11031, 1:15), and Alexa Fluor 488 goat anti-rat IgG (H + L) (Invitrogen Molecular Probes, A-11006, 1:15) for 1 hour. The coverslips were mounted with ProLong gold antifade reagent (Invitrogen, P36934) after a final PBS/CA630 rinse. Fluorescence microscopy was carried out using a Zeiss fluorescence microscope to monitor the IdU/CIdU nucleoside.

## Comet assay

Cells were collected by centrifugation at 150 g for 5 min. Samples were then washed once with 1X PBS and centrifuged again at 150 g for 3 min. Samples were then resuspended in 1 mL of 1X PBS at $1 \times 10^6$ cells/mL. On slides coated with 1% normal melting point agarose, a mixture of 1:7 cell suspension to 1% low melting point agarose was added to each slide and covered with a coverslip. Slides were placed in 4 °C for thirty minutes. Cover slips were gently removed, and cells were placed in lysis buffer (5 M NaCl, 0.5 M EDTA, 10 mM Tris-HCl ph10, 1% Triton) for 90 min at 4 °C. Lysis buffer was removed and denaturation buffer (300 mM NaOH, 1 mM EDTA) was added for 30 min at 4 °C. Denaturation buffer was removed, and slides were placed into an electrophoresis unit containing 1X TBE buffer. Electrophoresis was run at 3 V/cm for 30 min at room temperature. Slides were removed from the electrophoresis unit and put into 2 washes with distilled water for 5 min each. Slides were then dehydrated using washes with ice-cold 70% ethanol 3 times for 5 min each. Slides were left to dry at room temperature and stained with 1X Vista Green dye. Images were acquired using 20X magnification on Zeiss Axio microscope. Data was analyzed using OpenComet plugin via ImageJ.

## Non-Denaturing ssDNA gap detection by immunofluorescence

Exponentially growing cells were labelled with 50 uM IdU for 48 hours before treatment with MRE11 inhibitor- Mirin, DNA2 inhibitor- C5 (HY-128729, MedChem express) and 2 mM hydroxyurea in the last 4 h prior to collection. Cells were then collected by centrifugation and washed with PBS (Phosphate Buffered Saline). Cells were seeded in glass slides

by Cytospin4 for 6 min at 150 g. Cells were fixed and permeabilized with 0.5% Triton X-100 and 4% paraformaldehyde in PBS at room temperature for 15 min. Fixed cells were then blocked with 3% BSA for 1 hour at room temperature. Cells were incubated with primary antibodies against pRPA - anti-pRPA (ser4, ser8) (Bethyl Lab A300-245A-M, 1:1000) at 4 °C overnight. Cells were then washed 3 times with PBS at room temperature, followed by incubation with antibodies against IdU for 1 h at room temperature. Cells were washed 3 times with PBS followed by secondary antibodies for 1 h at room temperature. Cells were then mounted with Prolong with DAPI.

## In vitro cytotoxicity assay

Olaparib (Selleck Chem) was dissolved in dimethylsulfoxide at a concentration of 10 mM and stored at −80 °C as single-use aliquots. Cultured ATLL cells were seeded at a density of 0.35 million/ml in their regular growth medium and exposed to various concentrations of the Olaparib (0.01, 1, 3, 10, 30, 100uM) for 72 h. Resazurin (R&D Systems) - based viability assays were used to determine the 50% inhibitory concentration (IC50) values of Olaparib.

## Mitotic DNA Synthesis (MiDAS)

Harvesting mitotic cells: To augment mitotic cells, asynchronous cells were treated with CDK1 inhibitor RO-3306 (Merck, SML0569) at a final concentration 7 μM for 7:30 h, or in the last 7:30 h of APH treatment as and when needed. The cells were then washed with pre-warmed Edu-containing medium (37 °C) for three times within 5 min before being released into pre-warmed fresh medium with Edu at a final concentration of 20 μM for 45 min–1 h. This allows the cells to advance into mitosis. The mitotic population is then analyzed. Next, the cells are collected and centrifuged at RT at 350 g/rcf for 5 min to form a pellet. They are then washed in pre-warmed fresh medium two times before resuspending them into DPBS (1 million cells/ml of DPBS). Further, cells are seeded on Corning- single frosted microslides using Cytospin 4 (Thermoscientific, 2948-081) for 6 min at 150 g at medium acceleration. Once cells are seeded, they are fixed and permeabilized for 15 min with 0.5% Triton in DPBS containing 4% formaldehyde. The slides are then washed twice with DPBS- to be used for EdU labelling and IF, or they can be stored at 4 °C for a maximum two weeks. EdU detection and IF staining for MIDAS: The cells were blocked using blocking buffer (3% BSA in 1X PBS) for 2 h at RT. To detect EdU Click-iT™ chemistry was used as per the manufacturer's instructions (Click-iT™ EdU Cell Proliferation Kit for Imaging, Alexa Fluor™ 594 dye). EdU detection was coupled with IF staining. The cells were incubated with primary antibody diluted in the blocking buffer at 4 °C overnight; followed by three washes 15 mins each with DPBS. Next, the cells were incubated with secondary antibodies diluted in the blocking buffer for 1 h at RT, followed by three washes, 15 mins each with DPBS. Slides were then mounted with Prolong gold containing DAPI (Invitrogen Prolong Gold anti fade reagent with DAPI, P36935). The slides were allowed to dry overnight before IF imaging. Images were captured using- 63 mm oil immersion lens, Zeiss Axio microscope. Primary and Secondary antibodies for MiDAS experiments: Primary antibodies used were FANCD2 (1:1000, NB100-182, Novus), 53BP1 (1:5000, S1778, Cell Signaling or NB100-304, Novus), Cyclin-A (1:500, sc-271682, Santa Cruz). Secondary antibodies used were Anti-rabbit IgG Fab2 Alexa Fluor ® 488 (1:20000, 4412S, Molecular probes), were Anti-rabbit IgG Fab2 Alexa Fluor ®594 (1:20000, 8889S, Molecular probes), Anti-mouse IgG Fab2 Alexa Fluor ® 488 (1:20000, 4408S, Molecular probes), Anti-mouse IgG Fab2 Alexa Fluor ® 594 (1:20000, 8890S, Molecular probes).

Analysis: Under-replicated DNA at CFS: Mitotic cells were labelled for EdU and IF staining was performed. FANCD2 twin foci were used as a marker of CFS location. Each mitotic cell was analyzed for EdU labelled FANCD2 twin foci. Each FANCD2 twin foci, showed 1–2 EdU foci suggesting DNA replication at under-replicated regions called CFS.

Inherited Nuclear lesions: Cells arrested at G2 using RO-3306 (Merck, SML0569)) at 7 μM final concentration were released and allowed to continue with cell cycle for 1 h. After fixing, blocking and IF staining, and imaging, the cells were classified into Cyclin A positive and Cyclin A negative population. Cyclin A negative cells stained for distinct 53BP1 foci are analyzed as G1 nuclear lesions.

Micronuclei Analysis: Chromatin fragment staining positive for DAPI, proximal to parent nucleus was analyzed as micronucleus. These micronuclei were stained for proteins that mark damaged regions as well as for sensing cytosolic DNA. The DAPI stained micronuclei that are stained for H2AX, 53BP1, FANCD2, and pRPA were considered indications of damage response.

## Flow cytometry

To assess cell cycle, exponentially growing cells were pulsed with 5-ethynyl-2′-deoxyuridine (EdU) one hour before collection. Cells were then fixed and stained using Click-iT® Plus EdU Imaging Kit (Cat.# C10637) and Propidium Iodide (Life Technologies-Cat.# P3566). Samples were then analyzed using Attune II and compiled in FlowJo.

To assess apoptosis, cells were treated with Rev1 (JHRE-06 HY-126214, Medchem Express), POLQ (ART558- HY-141520, Medchem Express), and PARP inhibitors (Olaparib- Medchem Express) before collection. Live staining was done using AnnexinV and Propidium iodide to identify apoptotic population using Dead Cell Apoptosis Kit with Annexin V Alexa Fluor 488 Propidium Iodide – Cat#V13241. A schematic for the gating strategy used has been included in the source data for the appropriate figures.

## RNA-sequencing

RNA was extracted using Trizol and RNeasy Mini (QIAGEN, 74106) kits. RNA library preparations and sequencing were done by ActiveMotif using Illumina TruSeq RNA Sample Preparation v2 Guide (Illumina), and next-generation sequencing were performed using the Illumina Next-Seq platform (Illumina).Bioinformatics analysis was done with Active Motif. Differential analysis was performed using DESeq2. Gene Ontology was performed using Metascape.

## Quantification and statistical analysis

All statistical calculations were performed using GraphPad Prism v.9.0. A two-sided Student's t-test was used to calculated p-values in GraphPad Prism v.9.0. All the experiments were generally conducted with at least three biological replicates. Differences were considered significant at p-values: * <0.03, ** <0.0021, *** <0.0002, **** <0.0001.

## Reporting summary

Further information on research design is available in the Nature Portfolio Reporting Summary linked to this article.

# Data availability

All RNA sequencing data discussed in this study have been deposited at NCBI's Gene Expression Omnibus under the GEO series accession number GSE303280. Source data are provided with this paper.

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

## Acknowledgements

We thank the Rutgers Cancer Institute, Immune Monitoring and Flow Cytometry Shared Resource, which is supported, in part, with funding from the NCI-CCSG P30CA072770-5920 for their support. We also thank Dr. Tzeh Keong Foo for his valuable discussions. The work was supported by an American Cancer Society pilot award to A.M, R00HL136870 to A.M, R01CA266847 to B.H.Y., K08CA245251 to A.D.D., R37CA286444 to A.D.D., and R01CA275187 to A.D.D., the V Foundation for Cancer Research to A.D.D., the Hyundai Hope on Wheels Foundation and the American Lebanese Syrian Associated Charities to A.D.D., ALSF to J.Q., Curing Kids Cancer to J.Q., Wong Family foundation grant to J.Q., R35GM152228 to J.G. R01ES034733 to J.G., R01CA138804 to B.X., R01CA262227 to B.X., and P01250957-9485 to B.X.

## Author contributions

The project was conceived by A.M. Most of the experiments were conducted by A.B.G. and M.N. A.B.G. and M.N. contributed equally to this manuscript and hence share co-first authorship. The Comet assay was conducted by J.E.G.; Some of the EP300 and CBP immunoblots were carried out by A.D.D.; Data analysis was carried out by A.B.G., M.N., J.E.G., R.Z., V.K., A.J., A.P., P.M., and I.S. The NA-ATLL and J-ATLL cell lines were provided by B.H.Y. The JQAD1 compound was provided by J.Q. The data was analyzed and discussed by A.M., J.G., J.C., K.K., A.D.D., P.M., C.M., and B.H.Y. The manuscript was written by A.M.

## Competing interests

The authors declare no competing interests.
