## [Transparent Peer Review file · Nature Communications]

EP300 deficiency leads to chronic replication stress mediated by defective replication fork protection

Corresponding Author: Dr Advaita Madireddy

Version 0:

Reviewer comments:

Reviewer #1

(Remarks to the Author)

The authors examine the functional consequences of loss of p300, a major histone acetyltransferase. Using a panel of EP300 mutant cell lines and a specific PROTAC, they observed an S-phase delay, replication stress, DNA damage and elevated cell death in cells lacking p300 function. They next analyzed replication dynamics in the FRA6E region using SMARD and found that cells lacking p300 had more stalled forks and dormant origin firing, suggesting replication fork defects. Fiber analysis revealed that forks were able to stall in response to HU but were deprotected, leading to degradation and impaired restart, as well as the accumulation of ssDNA gaps. Consistent with this, treatment with the MRE11 inhibitor mirin partially rescued the phenotype and p300 deficient cells had fewer HR markers and reduced BRCA2 protein levels. They next examined markers of under-replicated DNA, as would be expected from the previous data. Increased FANCD2 twin foci, 53BP1 foci and p-RPA+ micronuclei were observed but not an increase in MiDAS. Micronuclei are a major trigger of the cGAS pathway and they find reduced cGAS levels in some of the mutant cells, despite higher levels of cytosolic DNA.

The analysis of phenotypes in cells deficient for p300 in isolation is overdue- as the authors point out, it has typically been studied in situations where CBP is also compromised. In this regard, the work will be of wide interest to the transcription/epigenetics and DNA damage fields. The manuscript is well organized and for the most part easy to follow with appropriate experiments used to analyze the phenotypes. There is an inherent amount of variability between some of the cell lines, making it unclear how inter-related some of the phenotypes are. The use of the PROTAC is a nice complement to the cell line mutants, and in some ways a better approach, given that each mutant cell line is going to have a host of other genetic alterations that may affect many of the outcomes. The paper makes the case that EP300 mutant ATLL cells parallel BRCA deficiency, and while I agree in some aspects, I am somewhat confused about what is going on mechanistically (see specific comments below). I also think there needs to be some clarity on biological repetition of experiments in the figure legends, where cell numbers analyzed are given in place of biological replicate n.

Comments

1. In figure 1, there is a bit of selectivity in the analysis that raises some concern about the consistency of the phenotype to the genotype. In the comet assays, there is little difference between J1 and NA4, while NA3 has more breaks. Then when gH2AX is assessed, NA2, that was not in the comet assay, is shown with NA3. So, is it fair to say that 1 (J1) of the 3 EP300 mutant cells does not show elevated DSBs? This becomes important in Figures 2/3, as J1 has perturbed fork progression compared to NA4. Is the break phenotype and fork progression phenotype separable? There does not seem to be a clear 1:1 relationship based on the data presented.
2. I think that some clarification on the n for some experiments is needed. For example, in Figure 3B (fibers), 4I or 4J (foci counts), an n of >100 is given, I assume referring to the number of cells scored. But how many times was the experiment itself actually performed? Are the scatterplots a mix of multiple repetitions or it was done once?
3. The authors explain the replication defects primarily through the observation that BRCA2 expression is impaired. There needs to be some precision in how the term expression is used for clarity. It is unclear from the text whether gene expression or steady state protein levels are being assayed in Figure 4H for example. How consistent is the reduction in BRCA2 protein levels among the other EP300 mutant cell lines, as only 1 is shown? Is it transcriptionally or post-transcriptionally reduced? The model hinges on this data to a large extent and it needs some clarification.
4. Given that BRCA2 expression and protein levels are modified by cell cycle and DNA damaging agents, how is the cell cycle affected by the PROTAC over the time course shown?
5. The authors examine cytosolic DNA, using gH2AX as a marker. They then say they confirm this with the "accumulation of

mislocalized ssDNA foci". Histones like H2AX are very unlikely to be on ssDNA, so I am confused about the proposed relationship between these endpoints and also how they discriminate between cytosolic ssDNA from the nucleus and mtDNA that also has ssDNA.

6. The cGAS data is difficult to follow. In Figure 6A, there are 4 WT cells and 1 has less cGAS. There are 3 mutants and 2 have low cGAS and the other is comparable to the WT with low cGAS. So, the mutants trend towards less, but it is not that clear- I understand that having a large number of cell lines is difficult. Then, using the PROTAC, cGAS mobility is increased after 72 hours but levels do not really change from what I can gather- what the red boxes are indicating is not explained. In 6E, the term expression is used without clarifying what was measured. This is mRNA or the protein in the previous blot? Given the clear discrepancies between the mutant cells where cGAS protein levels are variable and the PROTAC where there is a size change, I think the conclusion that p300 loss suppresses cGAS expression is not very well founded and no endpoints actually measuring its mRNA levels or innate immune output (IFN production, antiviral gene programs) are actually measured.

7. Given the prominent role for p300 in transcription, how do the gene expression patterns differ upon p300 depletion with the PROTAC? Is the gene expression of HR or innate immune genes impacted by p300 loss? This would help clarify point 6.

8. Given the suggestion that these cells closely resemble BRCA-deficient tumors, are they sensitive to PARP or ATR inhibitors?

Minor issues:

1. In the last sentence of first intro and a few other places in the manuscript, CBP is referred to as CPB.

2. In Figure 1G (and 4H), I would not say relative expression. This does not measure the expression of anything, it measures the steady state level of protein and/or phosphorylation. Are phosphorylation levels normalized to the total protein in the case of RPA and CHK1? They should be, instead of using Vinculin. I think it is worth clarifying how this was done, the graph in 1G for p-RPA in NA3 does not match the data shown as it goes up to equivalent levels at late time points in the graph and not in the blot.

Reviewer #2

(Remarks to the Author)

The paper "Acetyl transferase EP300 deficiency leads to chronic replication stress mediated by defective fork protection at stalled replication forks" by Barreto-Galvez et al., presents a comprehensive study on the role of EP300, an epigenetic regulator and transcriptional activator, in the context of adult T-cell leukemia/lymphoma (ATLL). The study utilizes ATLL cells to explore the effects of EP300 mutations, revealing that such mutations lead to extensive genomic instability, characterized by prolonged cell cycle kinetics, increased replication origin firing, and extensive nucleolytic degradation of nascent DNA at stalled replication forks. This results in the accumulation of single-stranded DNA gaps and a dependency on error-prone replication restart mechanisms. The research highlights that EP300 deficiency in ATLL cells leads to under-replicated DNA during the G2/M phase of the cell cycle, resulting in mitotic DNA synthesis associated with mitotic segregation defects, micronuclei formation, and the transmission of unrepaired inherited DNA lesions into the subsequent G1 phase. Additionally, EP300-deficient cells demonstrate a significant defect in innate immune system activation, suggesting a mechanism for immune evasion in aggressive cancers. Overall, the study is a significant contribution to understanding the molecular pathology of EP300 mutations.

Major concerns:

1. The paper provides valuable insights into potential mechanisms underlying EP300mut-related tumorigenesis. However, the manuscript does not support whether and how this knowledge may have any implications for cancer therapy.

2. The authors state that EP300-deficient cells demonstrate a significant defect in innate immune system activation, suggesting a mechanism for immune evasion in aggressive cancers. The manuscript lacks any relevant *in vitro* and importantly, *in vivo* experiments supporting this statement. The conclusion was formed based on the results presented in Fig 6C-E - lowered levels of cGAS in EP300 deficient or EP300 depleted cells- this represents an overstatement.

3. The authors state that EP300mut cells closely resemble BRCA-deficient cells. More rigorous approaches would be required to support this conclusion. Regardless, the manuscript does not address if, for example, EP300mut samples could benefit from treatments designed for BRCA deficient cells.

4. Mre11, DNA2, and Exo1 are the most common nucleases involved in nucleolytic processing of nascent DNA. Why the authors have focused on Mre11 only and did not test the remaining nucleases is unclear.

5. The studies with Mirin, a Mre11 inhibitor, should be complemented with molecular approaches (shRNAs or relevant).

6. The potential mechanistic link between EP300 depletion and BRCA2 downregulation is intriguing but it was not explored or discussed in more detail.

7. The authors mention Figure 7 in the manuscript narrative. Fig 7 itself is not included in the merged file. Conversely, there is no comment regarding Fig 3A in the manuscript narrative.

8. The discussion is not well developed.

9. Lastly, the study's reliance on cell line models may not fully capture the complexity of EP300 mutations in diverse clinical settings. This should be clarified.

Minor concerns:

Fig 1 - the exemplary images from the comet assay should be included.

Reviewer #3

(Remarks to the Author)

The manuscript entitled: "Acetyl transferase EP300 deficiency leads to chronic replication stress mediated by defective fork protection at stalled replication forks" attempts to show "that mutations in EP300 cause chronic DNA replication stress, persistent genomic instability and innate immune system evasion underlie aggressive chemo-resistant tumorigenesis in humans."

While many of the experiments that make up this manuscript are consistent with this premise there are several issues that need to be clarified before any conclusions concerning EP300 acetyl transferase activity and any of these genomic instability phenotypes can be drawn. The primary concern for this reviewer is in the specificity of the assays. Are the phenotypes that they describe due to mutation/degradation of EP300?

I recommend a major revision of this manuscript prior to publication.

First of all, the patient derived cell lines need to be further characterized or clarified. What is the nature of the mutations in EP300 in the ATLL cell lines? If they are loss of function, are they heterozygous or homozygous? If they are gain of function, do they also effect CBP activity? Is there any residual EP300 or CBP HAT or KAT activity? In addition, a comparison of the EP300 protein levels, i.e Western Blots, for the various ATLL cell lines should be included in the manuscript.

The mutational status of EP300 is an important issue in the ATLL lines. For example, the interpretation of the H2AX data following JQAD1 exposure in Figure 1E is incomplete, and therefore maybe misleading. The only data shown with JQAD1 is on a EP300 WT cells (NA4). What are the consequences of JQAD1 exposure on the EP300 mutant cells (NA2 and NA3)? If the mutations in EP300 are equivalent to a null mutation, then the prediction is that there would be no effect of JQAD1 exposure on gamma-H2AX foci in the EP300 mutant cells. This is an important control, as they state that: "We decided to proceed with the lower dose to minimize any potential toxicity." The analysis of the NA2 and NA3 (EP300 mutant) cells exposed to JQAD1 may allow them to determine if there is or is not any toxicity in the absence of EP300.

While the replication defects in JQAD1 treated cells are clearly documented, an additional control experiment is needed to conclude that the JQAD1 experiments are specific for EP300, and not due to off target effects. Inhibitor/degrader studies are always subject to the criticism of specificity. Are the affects that they observe specific to the protein that they target? Other labs (Stauffer et al for example) had previously used rescue of EP300 expression in siRNA treated cells, thus expression of siRNA resistant EP300 rescued the replication checkpoint defect in EP300 siRNA treated cells, thus demonstrating that loss of EP300 was indeed responsible for the checkpoint defects. Perhaps EP300 siRNA treated cells, with rescue experiments included, could answer this question and should give results consistent with the jQAD1 assays.

The data showing the analysis of replication dynamics within FRA6E in Figure 2 is quite convincing, and indicates that there is a clear difference between the ATLL cell lines. However, as pointed out above, the nature of the mutations in these cell lines needs to be addressed. Also, do the dormant origins fire in EP300 WT cells when exposed to JQAD1?

Minor points:

Figure 1B: it is not clear what cells were used for the panel marked "CONTROL"

Figure 1G: The legend says the cells were collected at 8 timepoints, which is what the Western Blots show, but the bar graphs only have 7 (no 32 h timepoint).

The labels for Figure 4A and 4B are confusing, is this a simple mix up of NA3 (EP300WT) and NA4 (EP300 Mut)?

Version 1:

Reviewer comments:

Reviewer #1

(Remarks to the Author)

The revised manuscript addresses the majority of the reviewer comments well. The authors have added additional new data that supports many of the points and extended the discussion significantly. I have a few remaining concerns that should be considered.

1. Please put the drug concentrations and specific drug used in the figure legends in all cases. The authors use the PARP inhibitor sensitivity data to make the point that these cells show similar phenotypes to BRCA deficient cells. The dose is not provided and only a single dose is used, making any sort of comparison difficult. While I agree they are more sensitive, the effect is quite mild (appears less than 2-fold). Compared to any BRCA deficient line I have tested, this is not very impressive, but without dosage info, it is somewhat hard to assess. I assume clonogenic assays are not feasible here due to the cell type, but a dose response would add confidence to this point. As this is also used to suggest a clinical application in the discussion, I am not sure a ~7% killing in 1 cell line would be a strong impetus for a clinical trial. Has anything been done with the Protac for example?

2. I wanted to look at the gene level data for the RNAseq, as there are a number of genes not highlighted here that can confer some BRCA-like phenotypes. However, it does not appear to be provided and the Supp panel data in the excel sheet

does not correspond to the new panel. This should be updated unless I missed something.

Reviewer #2

(Remarks to the Author)

he authors have addressed all concerns to my satisfaction.

Reviewer #3

(Remarks to the Author)

The authors have addressed all of my concerns. While some of the results concerning variability in phenotypes between different samples are still present, the results are consistent with their conclusions. I recommend publication in its current form.

Point-by-point response to the reviewers' comments, reproduced verbatim

We thank the reviewers for the extremely thoughtful and valuable comments. We have taken every question and comment to heart and have spent a lot of time and resources over the course of this revision period to address them to the best of our ability.

Response to Reviewer 1:

The authors examine the functional consequences of loss of p300, a major histone acetyltransferase. Using a panel of EP300 mutant cell lines and a specific PROTAC, they observed an S-phase delay, replication stress, DNA damage and elevated cell death in cells lacking p300 function. They next analyzed replication dynamics in the FRA6E region using SMARD and found that cells lacking p300 had more stalled forks and dormant origin firing, suggesting replication fork defects. Fiber analysis revealed that forks were able to stall in response to HU but were deprotected, leading to degradation and impaired restart, as well as the accumulation of ssDNA gaps. Consistent with this, treatment with the MRE11 inhibitor mirin partially rescued the phenotype and p300 deficient cells had fewer HR markers and reduced BRCA2 protein levels. They next examined markers of under-replicated DNA, as would be expected from the previous data. Increased FANCD2 twin foci, 53BP1 foci and p-RPA+ micronuclei were observed but not an increase in MiDAS. Micronuclei are a major trigger of the cGAS pathway and they find reduced cGAS levels in some of the mutant cells, despite higher levels of cytosolic DNA.

The analysis of phenotypes in cells deficient for p300 in isolation is overdue- as the authors point out, it has typically been studied in situations where CBP is also compromised. In this regard, the work will be of wide interest to the transcription/epigenetics and DNA damage fields. The manuscript is well organized and for the most part easy to follow with appropriate experiments used to analyze the phenotypes.

Response: We thank the reviewer for recognizing the impact and novelty of our work.

There is an inherent amount of variability between some of the cell lines, making it unclear how inter-related some of the phenotypes are. The use of the PROTAC is a nice complement to the cell line mutants, and in some ways a better approach, given that each mutant cell line is going to have a host of other genetic alterations that may affect many of the outcomes. The paper makes the case that EP300 mutant ATLL cells parallel BRCA deficiency, and while I agree in some aspects, I am somewhat confused about what is going on mechanistically (see specific comments below). I also think there needs to be some clarity on biological repetition of experiments in the figure legends, where cell numbers analyzed are given in place of biological replicate n.

Response: We thank the reviewer for recognizing our efforts to minimize ambiguity in data interpretation due to possible inter-cell line variability by our utilization of the EP300-specific PROTAC degrader. We have now performed several additional experiments that we feel

mechanistically strengthen our manuscript. Moreover, we have now repeated several experiments to improve scientific rigor which further solidifies our conclusions.

1. In figure 1, there is a bit of selectivity in the analysis that raises some concern about the consistency of the phenotype to the genotype. In the comet assays, there is little difference between J1 and NA4, while NA3 has more breaks. Then when gH2AX is assessed, NA2, that was not in the comet assay, is shown with NA3. So, is it fair to say that 1 (J1) of the 3 EP300 mutant cells does not show elevated DSBs? This becomes important in Figures 2/3, as J1 has perturbed fork progression compared to NA4. Is the break phenotype and fork progression phenotype separable? There does not seem to be a clear 1:1 relationship based on the data presented.

Response: To address this, we have repeated the phospho-H2AX Immunofluorescence assay as well as the Comet assay simultaneously in the same cell lines for consistency in the presence or absence of replication stress (aphidicolin; **New Figures 1D-E**). In addition, we have also shown the effect of PROTAC-mediated degradation of EP300 on DNA damage accumulation independent of exogenous replicative inhibition on the NA-ATLL cell lines irrespective of the EP300 status (**New Supp. Fig. 1G-J**).

Regarding the comparison between the Japanese cell lines EP300WT (J2) and EP300MUT (J1). We would like to apologize for the confusion. Our intention was not to compare the cell lines of Japanese origin to the cell lines of North American origin. What we instead meant to do was compare within a geographic area what the presence or absence of EP300 would do to the cells. While we expect the biology underlying EP300's absence in any given genetic background to have a similar phenotypic impact on the cells, we don't believe that the magnitude or extent of the phenotype observed to be comparable. We recognize that including both in the same histogram automatically warrants comparison of one with the other which doesn't seem scientifically accurate considering the numerous possible differences between the J and NA cell lines. The most basic example of this statement is that the cell lines of Japanese origin have been around for decades, while the NA-ATLL lines were generated in the last 5 years. To address this point, we have hence separated the J-ATLL data from the NA-ATLL data and have presented them as independent histograms that show similar trends in terms of EP300's influence on the different endpoints studied (**New Figure 3B; New Supp. Fig. 3A**).

2. I think that some clarification on the n for some experiments is needed. For example, in Figure 3B (fibers), 4I or 4J (foci counts), an n of >100 is given, I assume referring to the number of cells scored. But how many times was the experiment itself actually performed? Are the scatterplots a mix of multiple repetitions or it was done once?

Response: We have addressed this by clearly logging the number of replicates and number of cells independently. We also clearly state when the scatterplots present combined data points from multiple repetitions (three independent replicates are indicated by n=3. In most if not all the data presented, the scatterplots represent data combined from two if not

three replicates from independent cultures of cells. In the case of bar graphs, each experimental replicate's mean is presented as a dot (e.g. **New Figure 1H**).

3. The authors explain the replication defects primarily through the observation that BRCA2 expression is impaired. There needs to be some precision in how the term expression is used for clarity. It is unclear from the text whether gene expression or steady state protein levels are being assayed in Figure 4H for example. How consistent is the reduction in BRCA2 protein levels among the other EP300 mutant cell lines, as only 1 is shown? Is it transcriptionally or post-transcriptionally reduced? The model hinges on this data to a large extent and it needs some clarification.

Response: In the figures, we have clearly specified BRCA2 gene (mRNA) expression (**New Fig. 4G**) versus relative protein levels (**New Fig. 4A-D**).

To address the question of BRCA2 protein levels among the other EP300 mutant cell lines, we believe BRCA2 protein expression to be very low in general in all ATLL cell lines to begin with and this is evident from the faint bands that are barely detectable in Old Figures 4F-G in the EP300WT lanes. In addition, BRCA2 is a ~390kDa protein and is very difficult to blot. Despite multiple efforts by several individuals independently (Durbin, Niljekar, Barreto-Galvez and others), it has been very challenging to get a detectable band for all cell lines in a single blot. In this revision, we have presented two new blots that show BRCA2 levels in the NA1 EP300WT and NA2 EP300Mut cell lines that has not been presented in the previously version of the manuscript (**New Fig. 4C**). The new data demonstrate lower levels of BRCA2 protein expression in the NA2 EP300Mut line as compared to the NA1 EP300WT, and also the time-dependent loss of BRCA2 associated with PROTAC-mediated EP300 degradation in both NA1 and NA4 EP300WT lines but not the EP300Mut NA2 line (**New Fig. 4B-C**). These data suggest that EP300 downregulation is associated with lower BRCA2 protein expression.

To determine whether BRCA2 is transcriptionally or post-transcriptionally reduced, we performed an RNA-seq experiment (**New Fig. 4G; New Supp. Fig. 4D**). In the volcano plot presented that summarizes the RNA-seq data, we have labelled relevant genes involved in the replication fork protection pathway as those with significantly elevated gene expression (red dots) and those that are relatively unchanged or moderately upregulated gene expression (orange dots) between the EP300Mut and EP300WT cells. In particular, we have clearly marked BRCA2 gene expression (light orange dot), which falls within this range between the EP300Mut and WT cells. Importantly, there was no detectable decrease in the transcript levels of the genes involved in this mechanism, including BRCA2. This was confirmed by a gene ontology analysis of genes whose expression remained relatively unchanged by EP300's absence, where most of these genes were involved in DNA repair, cell cycle, homologous recombination (fork protection) and the mitotic cell cycle (**New Supp. Fig. 4D**). This data suggests that the variation in BRCA2 protein expression is post-transcriptionally reduced. Since the immunoblots assessing BRCA2 protein expression show a reduction and not complete absence of BRCA2 protein in EP300Mut ATLL cells, we

wanted to determine whether recruitment of BRCA2 protein to sites of replication stalling was affected in any significant way which could explain the replication defects clearly observed in the EP300Mut cells. The data assessing BRCA2 nuclear foci formation clearly shows a significant decrease in BRCA2 foci formation in EP300Mut NA-ATLL cells as compared to the wildtype cells (**New Fig. 4E**). Next, to confirm a functional decline in BRCA2's role in fork protection, we assessed RAD51 focus formation and found that lower BRCA2 foci observed in EP300Mut cells correlated with reduced RAD51 nuclear foci in these cells as compared to wildtype cells both in the presence and absence of replication stress (**New Fig. 4F**). These new data collectively substantiate functional loss of BRCA2 in EP300Mut ATLL cells.

4. Given that BRCA2 expression and protein levels are modified by cell cycle and DNA damaging agents, how is the cell cycle affected by the PROTAC over the time course shown?

Response: We agree that BRCA2 expression increases with cell cycle progression of cells into the S-phase and further increases upon replication stress. When cells are resting in G0/G1, BRCA2 expression is expected to be low. To address, therefore, if the cell cycle is changed by the PROTAC, we measured cell cycle by Propidium Iodide staining after PROTAC treatment over the time course utilized in the manuscript for the different assays. The results show that after PROTAC treatment, while the S phase fraction doubles in the EP300WT cells (NA1 and NA2), indicative of S-phase arrest likely due to replication stress, there is no prominent increase in the G0 population (**New Supp. Fig. 1G**). Importantly, based on the arrest observed, we should expect to see an increase in BRCA2 expression, however, we see a prominent decrease of BRCA2 protein levels after PROTAC treatment over time (**New Fig. 4B-C**) demonstrating that BRCA2 protein expression changes are not due to changes in the cell cycle. Furthermore, in the EP300Mut cells (NA2 and NA3), we see no significant difference in the distribution of cells across different phases of the cell cycle in response to PROTAC treatment showing that it's the EP300 loss but not the S-phase arrest observed in NA-ATLL cells that led to decreased BRCA2 protein (**New Supp. Fig. 1G**). To further clarify this point, we were very careful to only report BRCA2 and RAD51 recruitment/ foci formation in EdU-positive cells which is indicative of actively replicating cells (**New Fig. 4E-F**), to address this very important point raised by the reviewer.

5-6. All reviewer comments regarding Cytosolic DNA and cGAS data presented in Figure 6A-E:

Response: We had a discussion with our editor, Dr. Nele Hug regarding this point and we would like to remove the data pertinent to Cytosolic DNA and cGAS inactivation presented in **Old Figure 6A-E**. While these are very interesting findings with a strong potential toward identifying a novel therapeutic regimen in the future, based on all 3 reviewer comments, we recognize and agree that the data presented is preliminary and needs significant amount of further substantiation. In addition, we do feel these findings need more in-depth investigation which is beyond the scope of this manuscript. The main focus of this

manuscript is the novel discovery that EP300Mut ATLL cells closely resembling BRCA-deficient cells in all the endpoints measured. We have performed a substantial number of experiments and have incorporated further data to strengthen this aspect of our study in **(New Fig. 4)**.

7. Given the prominent role for p300 in transcription, how do the gene expression patterns differ upon p300 depletion with the PROTAC? Is the gene expression of HR or innate immune genes impacted by p300 loss? This would help clarify point 6.

Response: To address this point, we performed an RNA-seq analysis. While EP300 depletion with the PROTAC resulted in an overall striking change in the transcriptional landscape (**New Supp. Fig. 1**), most of the genes involved in HR were either moderately upregulated or remained unaltered transcriptionally by EP300 loss (**New Fig. 4G; New Supp. Fig. 4D**). This data combined with our immunoblots assessing protein expression of key fork protection/HR genes suggests post-transcriptional regulation of these proteins.

8. Given the suggestion that these cells closely resemble BRCA-deficient tumors, are they sensitive to PARP or ATR inhibitors?

Response: We evaluated sensitivity to ATRi, Wee1i, PARPi, REV1i and POLQi as part of this revision. While the EP300Mut cells showed moderate sensitivity to ATRi and Wee1i and a mild synergy upon combining ATRi and Wee1i, the data was not compelling. For this reason, we are not including this data in the manuscript. In the resubmitted manuscript, we have now presented data in **New Figure 4** to show that EP300Mut ATLL cells are highly sensitive to PARP inhibition and are also responsive to both Rev1 inhibition and PolQ inhibition. Importantly, the cells show an additive effect when Rev1i and PARPi are combined (**New Fig. 4I-J; New Supp. Fig. 4K**) clearly demonstrating the similarities between EP300-deficient and BRCA-deficient tumors.

Minor Comments:

1. In the last sentence of first intro and a few other places in the manuscript, CBP is referred to as CPB.

Response: This has been rectified.

2. In Figure 1G (and 4H), I would not say relative expression. This does not measure the expression of anything, it measures the steady state level of protein and/or phosphorylation. Are phosphorylation levels normalized to the total protein in the case of RPA and CHK1? They should be, instead of using Vinculin. I think it is worth clarifying how this was done, the graph in 1G for p-RPA in NA3 does not match the data shown as it goes up to equivalent levels at late time points in the graph and not in the blot.

Response: This has been addressed. We have now quantified pRPA levels relative to the total RPA levels in each cell lines (refer **New Fig. 1G**).

Response to Reviewer 2:

The paper "Acetyl transferase EP300 deficiency leads to chronic replication stress mediated by defective fork protection at stalled replication forks" by Barreto-Galvez et al., presents a comprehensive study on the role of EP300, an epigenetic regulator and transcriptional activator, in the context of adult T-cell leukemia/lymphoma (ATLL). The study utilizes ATLL cells to explore the effects of EP300 mutations, revealing that such mutations lead to extensive genomic instability, characterized by prolonged cell cycle kinetics, increased replication origin firing, and extensive nucleolytic degradation of nascent DNA at stalled replication forks. This results in the accumulation of single-stranded DNA gaps and a dependency on error-prone replication restart mechanisms. The research highlights that EP300 deficiency in ATLL cells leads to under-replicated DNA during the G2/M phase of the cell cycle, resulting in mitotic DNA synthesis associated with mitotic segregation defects, micronuclei formation, and the transmission of unrepaired inherited DNA lesions into the subsequent G1 phase. Additionally, EP300-deficient cells demonstrate a significant defect in innate immune system activation, suggesting a mechanism for immune evasion in aggressive cancers. Overall, the study is a significant contribution to understanding the molecular pathology of EP300 mutations.

Response:

1. The paper provides valuable insights into potential mechanisms underlying EP300mut-related tumorigenesis. However, the manuscript does not support whether and how this knowledge may have any implications for cancer therapy.

Response: We thank the review for recognizing the importance of our findings to EP300-related tumorigenesis. We apologize for not addressing this in more detail in the initial submission. We have now presented data to show that therapies that are currently in clinical use and preclinical testing for BRCA-deficient cancers have the potential to be effective in conferring sensitivity in EP300 Mut NA-ATLL cells (**New Fig. 4I-J; New Supp. Fig. 4K**). In addition, we have added a discussion section delving into the general implications of our findings for cancer therapy. Currently, there is an increasing interest in targeting EP300/CBP in cancer, with agents now in clinical trials (PMID: 37995682), and our own collaborators (Drs. Qi, Durbin) as well as many others are working to advance EP300-selective PROTACs to clinical trials.

2. The authors state that EP300-deficient cells demonstrate a significant defect in innate immune system activation, suggesting a mechanism for immune evasion in aggressive cancers. The manuscript lacks any relevant in vitro and importantly, in vivo experiments supporting this statement. The conclusion was formed based on the results presented in Fig 6C-E - lowered levels of cGAS in EP300 deficient or EP300 depleted cells- this represents an overstatement.

Response: We are in complete agreement with the stated reviewer comment. We had a discussion with our editor, Dr. Nele Hug regarding this point and we would like to remove the data pertinent to Cytosolic DNA and cGAS inactivation presented in **Old Figure 6A-E**. While these are very interesting findings with a strong potential toward identifying a novel therapeutic regimen in the future, we recognize and agree that the data presented is very preliminary and needs significant amount of substantiation. In addition, we do feel these findings need more in-depth investigation which is beyond the scope of this manuscript.

3. The authors state that EP300mut cells closely resemble BRCA-deficient cells. More rigorous approaches would be required to support this conclusion. Regardless, the manuscript does not address if, for example, EP300mut samples could benefit from treatments designed for BRCA deficient cells.

Response: We have now presented data in **New Fig. 4I-J** and **New Supp. Fig. 4K** to show that EP300Mut ATLL cells are highly sensitive to PARP inhibition and are also responsive to both Rev1 inhibition and PolQ inhibition. Importantly, the cells show a synergistic effect when Rev1i and PARPi are combined clearly demonstrating the similarities between EP300-deficient and BRCA-deficient tumors.

4. Mre11, DNA2, and Exo1 are the most common nucleases involved in nucleolytic processing of nascent DNA. Why the authors have focused on Mre11 only and did not test the remaining nucleases is unclear.

Response: According to literature, it is widely accepted that the nuclease employed to degrade nascent strands is largely dependent on the specific types of fork protection deficiency. For instance, it has been shown that MRE11 is needed for degradation in BRCA2-deficient cells (PMID: 21565612). In addition to this, another reason we focused on Mre11 was because it is the first nuclease that is involved in recognizing and initiating resection, while Exo1 and DNA2 contribute to extended resection. In any case we agree with the reviewer that it would be valuable to investigate their involvement. To this end, we performed numerous experiments aimed at si/shRNA depletion of all three nucleases that ended up being unsuccessful. The primary reason for this was that the NA-ATLL cells are very sensitive to cell culture manipulations, and they did not tolerate any of the transfection approaches we tried. The approaches we tried include lipofectamine, electroporation, and nucleofection and were all associated with extensive cell death in NA-ATLL cells. We utilize nucleofection and electroporation routinely in the lab especially since they yield better transfection efficiency than any other approach for suspension cells. Since DNA2 has an effective, commercially available inhibitor C5, we decided to proceed with this approach. The results from the DNA2 inhibition experiments recapitulate our finding from the Mre11 inhibition experiments and substantiate our findings (**New Fig. 3H**).

5. The studies with Mirin, a Mre11 inhibitor, should be complemented with molecular approaches (shRNAs or relevant).

Response: We tried our very best to perform this experiment and utilized a number of approaches all of which ended up being toxic to NA-ATLL cells since they are very sensitive to culture manipulations. We believe the newly presented data from our DNA2 inhibition studies, that recapitulate the Mre11 inhibition data (**New Fig. 3G**), provide confirmation of nuclease hyperactivity in mediating excessive single-stranded gap formation. The results consistently show that inhibition of these nucleases results in a significant reduction in the accumulation of ssDNA gaps in EP300Mut ATLL cells (**New Fig. 3H**).

6. The potential mechanistic link between EP300 depletion and BRCA2 downregulation is intriguing but it was not explored or discussed in more detail.

Response: We have explored this in much more detail in the re-submitted manuscript. To address the question of BRCA2 protein levels among the other EP300 mutant cell lines, we believe BRCA2 protein expression to be very low in general in all ATLL cell lines to begin with and this is evident from the faint bands that are barely detectable in Old Figures 4F-G in the EP300WT lanes. In addition, BRCA2 is a ~390kDa protein and is very difficult to blot. Despite multiple efforts by several individuals independently (Durbin, Niljekar, Barreto-Galvez and others), it has been very challenging to get a detectable band for all cell lines in a single blot. In this revision, we have presented two new blots that show BRCA2 levels in the NA1 EP300WT and NA2 EP300Mut cell lines that has not been presented in the previously version of the manuscript (**New Fig. 4C**). The new data demonstrate lower levels of BRCA2 protein expression in the NA2 EP300Mut line as compared to the NA1 EP300WT, and also the time-dependent loss of BRCA2 associated with PROTAC-mediated EP300 degradation in both NA1 and NA4 EP300WT lines but not the EP300Mut NA2 line (**New Fig. 4B-C**). These data suggest that EP300 downregulation is associated with lower BRCA2 protein expression.

To determine whether BRCA2 is transcriptionally or post-transcriptionally reduced, we performed an RNA-seq experiment (**New Fig. 4G; New Supp. Fig. 4D**). In the volcano plot presented that summarizes the RNA-seq data, we have labelled relevant genes involved in the replication fork protection pathway as those with significantly elevated gene expression (red dots) and those that are relatively unchanged or moderately upregulated gene expression (orange dots) between the EP300Mut and EP300WT cells. In particular, we have clearly marked BRCA2 gene expression (light orange dot), which falls within this range between the EP300Mut and WT cells. Importantly, there was no detectable decrease in the transcript levels of the genes involved in this mechanism, including BRCA2. This was confirmed by a gene ontology analysis of genes whose expression remained relatively unchanged by EP300's absence, where most of these genes were involved in DNA repair, cell cycle, homologous recombination (fork protection) and the mitotic cell cycle (**New Supp. Fig. 4D**). This data suggests that the variation in BRCA2 protein expression is post-transcriptionally reduced. Since the immunoblots assessing BRCA2 protein expression show a reduction and not complete absence of BRCA2 protein in EP300Mut ATLL cells, we wanted to determine whether chromatin recruitment of BRCA2 protein was affected in any

significant way which could explain the replication defects clearly observed in the EP300Mut cells. The data assessing BRCA2 nuclear foci formation clearly shows a significant decrease in BRCA2 chromatin recruitment in EP300Mut NA-ATLL cells as compared to the wildtype cells (**New Fig. 4E**). Next, to confirm a functional decline in BRCA2's role in fork protection, we assessed RAD51 focus formation and found that lower BRCA2 foci observed in EP300Mut cells correlated with reduced RAD51 nuclear foci in these cells as compared to wildtype cells both in the presence and absence of replication stress (**New Fig. 4F**). These new data collectively substantiate functional loss of BRCA2 in EP300Mut ATLL cells.

We discuss this in the discussion section of the manuscript.

7. The authors mention Figure 7 in the manuscript narrative. Fig 7 itself is not included in the merged file. Conversely, there is no comment regarding Fig 3A in the manuscript narrative.

Response: We apologize for the oversight and have clarified both points in the manuscript.

8. The discussion is not well developed.

Response: The discussion has been modified significantly in the form of tracked changes.

9. Lastly, the study's reliance on cell line models may not fully capture the complexity of EP300 mutations in diverse clinical settings. This should be clarified.

Response: This is a very valid point, and we have added an entire discussion section addressing this. In fact, our cell line data from geographically distinct locations (North American versus Japanese) suggests the ethnic diversity seen in the clinic and the differences in EP300Mut cell lines based on origin of these cell lines. These maybe very well applicable in the clinical setting in assessing PARP sensitivities.

Minor concerns:

1. Fig 1 - the exemplary images from the comet assay should be included.

Response: This has been included alongside New Supp. Fig. 1J.

Response to Reviewer 3:

The manuscript entitled: "Acetyl transferase EP300 deficiency leads to chronic replication stress mediated by defective fork protection at stalled replication forks" attempts to show "that mutations in EP300 cause chronic DNA replication stress, persistent genomic instability and innate immune system evasion underlie aggressive chemo-resistant tumorigenesis in humans." While many of the experiments that make up this manuscript

are consistent with this premise there are several issues that need to be clarified before any conclusions concerning EP300 acetyl transferase activity and any of these genomic instability phenotypes can be drawn. The primary concern for this reviewer is in the specificity of the assays. Are the phenotypes that they describe due to mutation/degradation of EP300? I recommend a major revision of this manuscript prior to publication.

Response:

1. First of all, the patient derived cell lines need to be further characterized or clarified. What is the nature of the mutations in EP300 in the ATLL cell lines? If they are loss of function, are they heterozygous or homozygous? If they are gain of function, do they also effect CBP activity? Is there any residual EP300 or CBP HAT or KAT activity?

Response: The spectrum of mutations observed in EP300Mut NA-ATLL cells include missense, splice site, and truncating mutations. These are heterozygous loss-of-function mutations, and the cell lines do not have any mutations in CBP. While the EP300Mut NA-ATLL cells we have analyzed in the paper show very low to absolutely no detectable p300 protein expression, the wildtype cells show a prominent p300 band as assessed by immunoblotting (**New Supp. Fig. 1A**). While this has been characterized in detail in Dr. Hilda Ye's (co-author) manuscript published in Blood (PMID: 30104217/), we have included a **Supplemental Table 1** that addresses this point. In addition, we find that the CBP protein is expressed in all NA-ATLL cells analyzed irrespective of the EP300 mutational status (**New Supp. Fig. 1A**). While we were able to demonstrate that the H3K27ac mark is reduced in PROTAC treated EP300WT cells (**New Supp. Fig. 1E**), we are unable to address the question regarding residual HAT or KAT activity since there are currently no tools available (commercially or through our EP300/CPB experts, Drs. Jun Qi and Dr. Adam Durbin) that will allow us to evaluate this.

2. In addition, a comparison of the EP300 protein levels, i.e Western Blots, for the various ATLL cell lines should be included in the manuscript.

Response: We have included a blot to address this point in **New Supplemental Figure 1A**. As expected, the EP300Mut NA-ATLL cells we have analyzed in the paper show very low to absolutely no detectable EP300 protein expression and the wildtype cells show a prominent EP300 band as assessed by immunoblotting.

3. The mutational status of EP300 is an important issue in the ATLL lines. For example, the interpretation of the H2AX data following JQAD1 exposure in Figure 1E is incomplete, and therefore maybe misleading. The only data shown with JQAD1 is on a EP300 WT cells (NA4). What are the consequences of JQAD1 exposure on the EP300 mutant cells (NA2 and NA3)? If the mutations in EP300 are equivalent to a null mutation, then the prediction is that there would be no effect of JQAD1 exposure on gamma-H2AX foci in the EP300 mutant cells. This is an important control, as they state that: "We decided to proceed with the lower dose to

minimize any potential toxicity.” The analysis of the NA2 and NA3 (EP300 mutant) cells exposed to JQAD1 may allow them to determine if there is or is not any toxicity in the absence of EP300.

Response: We agree completely with the reviewer and have now presented JQAD1 inhibition experiments for all the NA-ATLL cell lines assessing cell cycle kinetics, gamma-H2AX foci, and comet tails. As observed before, EP300Mut NA-ATLL cells show an inherent increase in the S-phase fraction, as compared to EP300WT cells, indicative of cell cycle arrest. However, in line with the reviewer’s prediction, our new data assessing the effect of JQAD1 treatment in EP300Mut NA-ATLL revealed no further significant increase in the percentage of cells in S-phase as determined by cell cycle profiling. In addition, there was no significant increase in the gamma-H2AX accumulation or comet tail movement in EP300Mut JQAD1-treated cells. In contrast, in the same set of experiments we observed a significant increase in the S-fraction in EP300WT cells treated with JQAD1, as compared to untreated cells. In addition, there is also a marked increase in DNA damage accumulation in EP300WT cells exposed to JQAD1. We have included the new data in **New Supplemental Figures 1G, 1H, 1I and 1K.**

4. While the replication defects in JQAD1 treated cells are clearly documented, an additional control experiment is needed to conclude that the JQAD1 experiments are specific for EP300, and not due to off target effects. Inhibitor/degrader studies are always subject to the criticism of specificity. Are the effects that they observe specific to the protein that they target? Other labs (Stauffer et al for example) had previously used rescue of EP300 expression in siRNA treated cells, thus expression of siRNA resistant EP300 rescued the replication checkpoint defect in EP300 siRNA treated cells, thus demonstrating that loss of EP300 was indeed responsible for the checkpoint defects. Perhaps EP300 siRNA treated cells, with rescue experiments included, could answer this question and should give results consistent with the JQAD1 assays.

Response: While we agree with the reviewer that a second approach such as EP300 siRNA treatment or the reconstitution of the wildtype EP300 protein would be very valuable in clarifying off-target effects of PROTAC degraders, these experiments have proved to be impossible to achieve at our end in ATLL cell lines. Our co-authors Dr. Durbin Dr. Qi have extensively assessed and characterized the off target of JQAD1 (PMID: 34772733). Our attempts at assessing this point have contributed significantly to our highly extended revision periods. Our siRNA treatments using an array of approaches proved to be too toxic for the cells which are very sensitive to culture condition changes. Simultaneously, we tried CRISPR-Cas9/RNP approaches to introduce similar mutations in the EP300 gene in wildtype and control cells which also proved to be unsuccessful. We also worked extensively with Dr. Adam Durbin, an expert on all things EP300, and ended up realizing that it realistically not possible to complement cells with wildtype p300 due to its large size. We believe that overexpression of EP300 in cells that have evolved to survive without it, likely makes it toxic to them. In the end, we decided to try a JQAD1 washout experiment to assess the cellular response to JQAD1 withdrawal and EP300 protein re-expression. In this

experiment, all NA-ATLL cells were exposed to JQAD1 for 48h and the untreated and JQAD1 treated cells were collected at 48h and gamma-H2AX foci formation was assessed by immunofluorescence. At the same time, a portion of the treated cells were released into PROTAC-free media and cultured for 72h to restore EP300 expression in the same culture of cells and the gamma-H2AX foci formation was measured in these cells at the 72h mark by immunofluorescence (**New Supp. Fig. 1I**). There were three things we observed from these set of experiments. First, EP300Mut NA-ATLL cells (NA2, NA3) as expected from our previous data (**New Supp. Fig. 1H**) inherently had more gamma-H2AX foci than EP300WT (NA1 and NA4) cells. Second, JQAD1 treatment as expected increased DNA damage in WT cells but did not in EP300Mut cells. Third, 72h after release from JQAD1 treatment (washout), the gamma-H2AX foci accumulation decreases significantly indicating that restoration of EP300 expression was likely mediating this effect. In contrast, no change in gamma-H2AX foci was observed after release from JQAD1 in EP300Mut cells.

5. The data showing the analysis of replication dynamics within FRA6E in Figure 2 is quite convincing and indicates that there is a clear difference between the ATLL cell lines. However, as pointed out above, the nature of the mutations in these cell lines needs to be addressed. Also, do the dormant origins fire in EP300 WT cells when exposed to JQAD1?

Response: We have included new SMARD profiles in the manuscript that clearly shows that EP300WT cells when treated with JQAD1 activate dormant replication origins as compared to untreated cells (**New Fig. 2F-G; New Supp. Fig. 2B**). Despite SMARD being highly labor intensive and time consuming, we were able to successfully address this reviewer concern in this revision.

Response:

Minor points:

1. Figure 1B: it is not clear what cells were used for the panel marked “CONTROL”

Response: The panel marked as CONTROL are non-ATLL transformed B-lymphocytes (marked B-Lymp in the resubmitted manuscript) that are EP300WT, that we routinely use in the lab. We have now clarified this in the manuscript.

2. Figure 1G: The legend says the cells were collected at 8 timepoints, which is what the Western Blots show, but the bar graphs only have 7 (no 32 h timepoint).

Response: This was a mistake that has now been corrected.

3. The labels for Figure 4A and 4B are confusing, is this a simple mix up of NA3 (EP300WT) and NA4 (EP300 Mut)?

Response: This has been addressed.

Point-by-point response to the reviewers' comments, reproduced verbatim

We thank Reviewer 1 for the extremely thoughtful and valuable comments. We have addressed each comment to the best of our ability. We also thank the other reviewers for their time in evaluating our revised manuscript.

Response to Reviewer 1:

The revised manuscript addresses the majority of the reviewer comments well. The authors have added additional new data that supports many of the points and extended the discussion significantly. I have a few remaining concerns that should be considered.

1. Please put the drug concentrations and specific drug used in the figure legends in all cases.

Response: We have now included all the drug concentrations and duration of treatments in the figure legends.

2. The authors use the PARP inhibitor sensitivity data to make the point that these cells show similar phenotypes to BRCA deficient cells. The dose is not provided and only a single dose is used, making any sort of comparison difficult. While I agree they are more sensitive, the effect is quite mild (appears less than 2-fold). Compared to any BRCA deficient line I have tested, this is not very impressive, but without dosage info, it is somewhat hard to assess. I assume clonogenic assays are not feasible here due to the cell type, but a dose response would add confidence to this point. As this is also used to suggest a clinical application in the discussion, I am not sure a ~7% killing in 1 cell line would be a strong impetus for a clinical trial. Has anything been done with the Protac for example?

Response: We thank the reviewer for this suggestion. To address this, we performed an invitro toxicity assay in our panel of EP300Mut and EP300WT NA-ATLL cell lines with varying concentrations of Olaparib. From this dose response data, we were able to establish an IC50 of ~30uM for EP300Mut NA-ATLL cell lines. The Olaparib IC50 range for the NA-ATLL cells (20-39.5 uM) is very comparable to that reported for Olaparib-sensitive breast cancer cell lines (10–60 uM) based on a 3-day cell viability assay. We have included this new data panel in New Supplementary Figure 4K. We believe this data supports our previous conclusion that NA-ATLL cells with mutated EP300 show similar phenotypes to BRCA deficient cells.

2. I wanted to look at the gene level data for the RNAseq, as there are a number of genes not highlighted here that can confer some BRCA-like phenotypes. However, it does not appear to be provided and the Supp panel data in the excel sheet does not correspond to the new panel. This should be updated unless I missed something.

Response: We thank the reviewer for pointing this out. We have now included a supplementary table (New Supplementary Table 2), that provides a list of all the differentially expressed genes involved in the replication fork protection pathway. The

highlighted genes that appear earlier in the table are the significantly up/down regulated genes. The entire list of genes from the RNA-seq data plotted in Figure 4G is available as part of the source data.